# Identification of Wheat *Septoria tritici* Resistance Genes in Wheat Germplasm Using Molecular Markers

**DOI:** 10.3390/plants13081113

**Published:** 2024-04-16

**Authors:** Alma Kokhmetova, Ardak Bolatbekova, Yuliya Zeleneva, Angelina Malysheva, Sholpan Bastaubayeva, Kanat Bakhytuly, Yerlan Dutbayev, Vladimir Tsygankov

**Affiliations:** 1Institute of Plant Biology and Biotechnology, Almaty 050040, Kazakhstan; ardashka1984@mail.ru (A.B.); malysheva_angelina@list.ru (A.M.); kanat1499@gmail.com (K.B.); 2All-Russian Research Institute of Plant Protection, Pushkin, St. Petersburg 196608, Russia; zelenewa@mail.ru; 3Kazakh Research Institute of Agriculture and Plant Growing, Almalybak 040909, Kazakhstan; sh.bastaubaeva@mail.ru; 4Department of Horticulture, Plant Protection and Quarantine, Faculty of Agrobiology, Kazakh National Agrarian Research University, Almaty 050000, Kazakhstan; edutbaev@mail.ru; 5Aktobe Agricultural Experimental Station, Aktobe 030014, Kazakhstan; zigan60@mail.ru

**Keywords:** wheat, *Zymoseptoria tritici*, adult plant resistance, seedling resistance, resistance genes, molecular markers

## Abstract

*Zymoseptoria tritici* (*Z. tritici*) is the main threat to global food security; it is a fungal disease that presents one of the most serious threats to wheat crops, causing severe yield losses worldwide, including in Kazakhstan. The pathogen leads to crop losses reaching from 15 to 50%. The objectives of this study were to (1) evaluate a wheat collection for *Z. tritici* resistance during the adult plant and seedling growth stages, (2) identify the sources of resistance genes that provide resistance to *Z. tritici* using molecular markers linked to *Stb* genes, and (3) identify potentially useful resistant wheat genotypes among cultivars and advanced breeding lines. This study evaluated 60 winter and spring wheat genotypes for *Z. tritici* resistance. According to the field reactions, 22 entries (35.7%) showed ≤10% disease severity in both years. The resistant reaction to a mix of *Z. tritici* isolates in the seedling stage was associated with adult plant resistance to disease in four wheat entries. The resistance of Rosinka 3 was due to the presence of *Stb8*; Omskaya 18 showed an immune reaction in the field and a moderately susceptible reaction in the seedling stage, possibly provided by a combination of the *Stb7* and *Stb2* genes. The high resistance in both the adult and seedling stages of Omskaya 29 and KR11-03 was due to the *Stb4* and *Stb2* genes and, possibly, due to the presence of unknown genes. A linked marker analysis revealed the presence of several *Stb* genes. The proportion of wheat entries with *Stb* genes was quite high at twenty-seven of the genotypes tested (45.0%), including four from Kazakhstan, nine from Russia, nine from the CIMMYT-ICARDA-IWWIP program, and five from the CIMMYT-SEPTMON nursery. Among the sixty entries, ten (16.7%) carried the resistance genes *Stb2* and *Stb8*, and the gene *Stb4* was found in seven cultivars (11.6%). Marker-assisted selection can be efficiently applied to develop wheat cultivars with effective *Stb* gene combinations that would directly assist in developing durable resistance in Kazakhstan. Resistant genotypes could also be used as improved parents in crossing programs to develop new wheat cultivars.

## 1. Introduction

Winter wheat (*Triticum aestivum* L.) is an important crop in the region of Central and West Asia (CWA), stretching from Kazakhstan and Afghanistan to Iran and Turkey. The crop is grown under both irrigated and rainfed conditions on ~13 Mha, and its average grain yield is ~2.5 t ha^−1^, which is far below the potential of 4–5 t ha^−1^ [1]. There are several reasons for poor wheat grain yield in the region, including changes in cultural practices including shifts from conventional tillage and stubble burning to reduced tillage practices, breaking of the technology of fertilizing, violation of or non-compliance with crop rotation, and shallow tillage and wheat monoculture involving the cultivation of susceptible cultivars. However, diseases and pests also play an important role in yield reduction [2].

Kazakhstan plays an important role in regional and global food security since most of the grain produced is sold in these regions. According to experts from the Food and Agriculture Organization of the United Nations (FAO), the world’s population will double by 2050. Ensuring food security is the most important priority of the economic strategy of Kazakhstan, as worsening the situation can deform the process of political and economic reforms and become a threat to the internal security of the state. World grain production has been increasing in recent years; at the same time, wheat losses from diseases account for about 10% of the potential crop around the world [3].

The widespread introduction of zero and minimal wheat cultivation technologies contributes to the development and harmfulness of leaf spot diseases (LSDs), the infection of which remains on crop residues. In the period 2000–2015, epiphytotic developments of leaf rust separately or together with Septoria occurred eight times in Kazakhstan [4]. *Zymoseptoria tritici* epidemics in the North of Kazakhstan occur five times every 10 years. According to the monitoring of *Z. tritici* in the Akmola region, there is a trend of increased development and severity of the disease, and, in recent years, the morbidity rate has reached a critical level. The disease manifested itself even during the years of hard drought (2003–2010). A strong development of *Z. tritici* was observed in 2013, 2014, and 2016 [5]. In Kazakhstan, five species of Septoria fungi were registered on wheat; the dominant species among them are *Parastagonospora nodorum* (Berk.) Quaedvl., Verkley et Crous and *Zymoseptoria tritici* (Desm.) Quaedvl. et Crous [4]. *Septoria tritici* blotch (STB) caused by the ascomycete fungus *Zimoseptoria tritici* is the major devasting foliar disease that causes significant yield loss in the wheat-growing regions of Kazakhstan. The life cycle of *Zymoseptoria tritici* in Kazakhstan is the same as in other regions. The life cycle of *Z. tritici* is broadly divided into two distinct stages, namely the symptomless latent phase and the necrotrophic stage. The latent phase can be further subdivided into three stages: transition, ingress, and colonization [4]. Yield losses of grain crops due to these two diseases amount to 9 million tons worldwide [6]. The fungus *P. nodorum* was discovered in 1960 in the Akmola region of Kazakhstan on soft wheat; since then, the disease has been observed in all regions of the country. *Septoria tritici* blotch and *Stagonospora* nodorum blotch are a major threat to global food security [7]. The average loss from pathogens can reach from 15 to 50% [4]. Leaf blotch diseases are a problem in wheat production in the Pacific Northwest and Northern Great Plains of the USA, Europe, and Central and West Asia [6,7,8,9,10,11,12]. Apart from leaf blotch diseases, leaf rust (*Puccinia triticina* Erikss.) [2,13,14], yellow rust (*Puccinia striiformis* Westend.) [15,16,17], and stem rust (*Puccinia graminis* f. *tritici* Erikss. & Henning) [18,19] are also widespread in Kazakhstan.

The inheritance of resistance to *Zymoseptoria tritici* (*Z. tritici*) can be either qualitative (specific for the isolate) or quantitative (non-specific for the isolate). Twenty-one qualitatively inherited major genes (*Stb1* to *Stb18*, *StbSm3*, *StbWW*, and *TmStb1*) have been detected hitherto and mapped on 14 wheat chromosomes against the STB pathogen [20]. These genes are generally effective but are genotype-specific, and their potency is only for a particular isolate of the pathogen [20]. Their act of resistance is supposed to be a gene-for-gene relationship. The majority of Kazakhstani wheat varieties encompass qualitative and quantitative genes, although several of the major genes are no longer effective in fields [4]. The *Phaeosphaeria nodorum* fungus produces eight necrotrophic effectors (NEs)/host-specific toxins (HSTs): SnToxA, SnTox1, SnTox2, SnTox3, SnTox4, SnTox5, SnTox6, and SnTox7 [21]. Each HST interacts with host sensitivity genes (Tsn1, Snn1, Snn2, Snn3, Snn4, Snn5, Snn6, and Snn7) [22]. Most qualitative genes are effective only against *Z. tritici* avirulent genotypes. Partial/quantitative resistance (QR) is more effective, which is effective against almost all pathogen genotypes [20]. There is information about the identified quantitative trait loci (QTLs) for QR Septoria [7,23]. QTLs are identified on chromosomes 6A, 7A, 3BS, 2D, 3A, 5B, 5A, 5B, 5D, 1BS (Snn1), 5BS (Snn3), 2DS (Snn2), 3AL, 4AL, 4BS, 5AS, 2DL, and 7AL [24,25,26]. In biparental mapping populations, 167 QTLs were detected [27]. All chromosomes except 5D carry at least one QTL or meta-QTL resistance to *Z. tritici* [20]. QTLs of seedling resistance to *Septoria* are located on chromosomes 1B, 2B, 2D, 3A, 4A, 4B, 5A, 5B, 5D, 6A, and 7A [26,28], and in adult plants, on chromosomes 2B, 2D, 4A, 4B, 5A, 6B, 7A, and 7B [29].

The disease reduces the assimilation surface of the leaves, causes underdevelopment of the ear, and significantly reduces the yield and quality of the grain [1]. To prevent losses, crops are treated with fungicides, which require additional costs. Comprehensive disease control strategies (cultivating resistant varieties, crop rotation) are the most effective, environmentally friendly, and economical means to combat wheat *Septoria*. Hence, identifying the sources of STB genes with minor-to-moderate effects will be used as an effective way to determine durable and resistant wheat varieties for STB. However, traditional breeding methods are not always effective, as the development of resistant varieties is a laborious and lengthy process. In this regard, the use of the marker-assisted selection (MAS) approach accelerates the cultivation of new cultivars. MAS breeding and the identification of target genes associated with disease resistance make the process of developing and introducing wheat varieties more accurate and reliable.

The objectives of this study were to (1) evaluate a wheat collection for *Zymoseptoria tritici* resistance during the adult plant and seedling growth stages, (2) identify the sources of resistance genes that provide resistance to *Z. tritici* using molecular markers linked to *Stb* genes, and (3) identify potentially useful resistant wheat genotypes among cultivars and advanced breeding lines.

## 2. Results

### Reaction of the Wheat Collection to Zymoseptoria tritici

Analysis of variance (ANOVA) revealed highly significant effects for wheat genotypes on resistance to *Zymoseptoria tritici* in both field and laboratory tests (Table 1). A total of 60 cultivars and breeding lines were analyzed over 2 years for field trials and 3 replicates for laboratory screening of seedling resistance. The resistance score (AUDPC, field/*Z. tritici*, seedling) to *Z. tritici* in ANOVA was taken as a variable; the year of field testing (n = 2, for APR) or replication (n = 5, for ASR) and genotype (n = 60) were taken as factors. There was significant variation among genotypes and disease severity (*p* < 0.001) in both growing seasons of field trials (*p* < 0.001).

The ANOVA results suggest that genotype is the predominant source of variation in resistance to *Z. tritici* in the collection, explaining 94.44% of the total variance in field trials and 74.24% of the variance in seedling resistance. All genotypes exhibited a high broad-sense heritability (0.94 and 0.74) for all the traits, indicating that resistance to *Z. tritici Septoria* can be improved by breeding (Table 1).

The resistance score with the AUDPC values for 2020 and 2021 for each genotype is presented in Figure 1. This clarifies whether the quantification is consistent across years. The Pearson correlation analysis revealed a significant positive correlation between the two years of field trials (r = 0.94; *p* < 0.001). No correlation was found between the resistance of adult plants and seedlings.

The genetic parameters of 60 winter wheat genotypes are presented in Table 2. Wide variability exists for *Z. tritici* resistance in the wheat collection for field and seedling resistance as evidenced by the presence of highly significant genotypic variance (Table 2). The CV ranged from 30.3% (*Z. tritici*, seedling) to 80% (AUDPC in 2020).

The distribution of accessions concerning *Z. tritici* infection types in the adult plant stage (a) and seedling stage (b) with an indication of the standard error is presented in Figure 2. Septoria development varied greatly among the wheat entries. Most of the entries showed higher levels of resistance in the adult stage under field conditions than as seedlings (Table 3, Figure 2). According to the field reactions in 2020, the majority of the studied genotypes (86.6%) had highly resistant (RR) and resistant (R) reactions to *Z. tritici;* in 2021, 49 wheat entries (81.6%) had RR and R reactions, indicating that these wheat entries are promising sources for adult plant resistance (Figure 2). Among the 60 genotypes, 8 (13%) and 11 (18.3%) wheat entries were susceptible in 2020 and 2021, respectively. The most susceptible cultivar over the two years was wheat cultivar Egemen, with AUDPC values of 507.5 and 595 in 2020 and 2021, respectively (Table 3). Twenty-two genotypes (Severyanka, Rosinka 3, Omskaya 18, Omskaya 28, Omskaya 29, Omskaya 35, Omskaya 36, Mironovskaya 808, Pamyati Azieva, KR11-40, KR11-03, NANJTNG82149/KAUZ, CROC 1AE. SQUARROSA, GAN/AE.437. SQUARROSA, TRAP#1/BOW, EFED/F5.83 7792(BAJAS), Batyr, Bayandy, Samgay, and Sapaly) showed ≤10% disease severity in both years and were considered as resistant in the adult plant stage under field conditions.

The seedling reactions of 60 wheat genotypes to the mix of *Z. tritici* isolates differed greatly. The wheat genotypes showed arrays of patterns in their responses to *Z. tritici* (Table 3). Most wheat cultivars and breeding lines were generally moderately susceptible (53.3%) and susceptible (16.6%) to *Z. tritici*. Of the 60 wheat entries, 18 had average disease reaction scores of less than two as seedlings, and 5 of these had reaction scores equal to one, indicating that these wheat entries were resistant. Eighteen wheat entries (Saratovskaya 42, Saratovskaya 55, Omskaya 29, Omskaya 35, KR11-9025, KR12-07, KR11-03, KR11-9014, P83-5112/V82274, NANJTNG82149/KAUZ, CROC_1/AE.SQUARROSA (205)//BORL95/3/2/*Milan, JAC161/TEMU51.80, BR14/CEP847-2, ECHA/LI115, Celinnaya 50, Manshyk, Sapaly, and Steklovidnaya 24) were resistant to *Z. tritici*. Reactions to *Z. tritici* isolates were associated with adult plant resistance to Septoria in four wheat entries, including Rosinka 3, Omskaya 18, Omskaya 29, and KR11-03. These genotypes were highly resistant both under greenhouse and field conditions.

The results of genotyping with markers linked to *Stb* resistance genes are given in Table 3. One or more *Stb* genes were detected in 23 of the 60 genotypes tested (38.3%) (Table 3). The molecular marker linked to *Stb2*, *WMS389*, amplified the fragment size of 150 bp in 10 genotypes, including Omskaya 18, Omskaya 35, Pamyati Azieva, KR11-03, KR12-10, KR11-9014, SOMO/SORA/ACTS5, EFED/22150, Egemen, and Sultan 2, while the remaining 50 genotypes were lacking *Stb2* (Table 3). In total, approximately 17% of all 60 entries assayed with SSR markers in this study were predicted to possess *Stb2*. As an example, the PCR results for 18 genotypes are shown in Figure 3. Five genotypes (Omskaya 19, SOMO/SORA/ACTS5, KR11-03, and Omskaya 38) had 150 bp fragments, indicative of the *Stb2* resistance gene. Twelve genotypes (Saratovskaya 70, KR12-5075, Batyr, Kazakhstanskaya 4, Severyanka, KR12-5001, Kyzylbiday, TOO11/TOOOO7, TALHUENJNJA, Botagoz, DOMOJNJA, F133/SHA5//OPATA, and Sultan 2) had no amplification product, indicative of the *Stb2* gene (Figure 3).

The presence of the *Stb4* gene in wheat genotypes was studied using the SSR marker *WMS111*. Of the 60 cultivars/lines identified to carry this resistance gene in our study, 7 genotypes (11.6%) amplified 210/220 bp fragments, indicating the presence of the *Stb4* resistance gene. These wheat entries include Albidum 31, Omskaya 29, KR11-13, KR12-18, KR11-26, TRAP#1/BOW, and EFED/F5.83-7792(BAJAS) (Table 3). There is no information confirming variations in the *Stb4* gene. Adhikari et al., 2004, showed that the *Stb4* locus from the wheat cv. Tadinia showed resistance to *Z. tritici* in both the seedling and adult plant stages [30]. Similar results were obtained in our studies. Based on the molecular screening results, all of the seven genotypes containing *Stb4* genes showed fairly good to moderate resistance responses during field testing. The *Stb4* gene had a large effect on *Z. tritici*, which provided a high level of field resistance in both years, with a final disease score of 5–10%. These genotypes, also exhibited seedling resistance values from moderately susceptible to susceptible.

SSR marker *WMS44* was used to screen the 60 wheat entries for *Stb5* gene detection. The expected size of the fragment amplification for locus *WMS44* coupled to the resistant allele of the *Stb4* gene was 182 bp. Out of 60 genotypes tested for *Stb5*, the expected PCR product was not amplified in any wheat entry.

Screening with the *WMC3133* marker produced the expected 197 bp band associated with the *Stb7* gene in four genotypes (6.7%), as well as in the control cultivar Estanzuela Federal (EFED/F5.83 7792(BAJAS), KR12-9012, Omskaya 18, and Omskaya 36) (Table 3). The other 56 wheat genotypes (93.3%) failed to amplify this gene.

Using the marker *WMS577*, the 180 bp fragment was amplified in the tested wheat entries. The molecular marker *WMS577*, linked to the *Stb8 gene*, amplified fragment size 180 bp in 10 genotypes (16.7%) (Saratovskaya 55, Rosinka 3, Mironovskaya 808, KR12-9011, KR12-5075, CROC_1/AE SQUARROSA (205)//BORL95/3/2/*Milan, EFED/22150, Egemen, Kokbiday, Kupava/Avocet (S)-1), while the remaining 50 genotypes were lacking *Stb7* (Table 3).

An analysis of the general distribution of dependent variables of various wheat samples characterized by the presence and absence of *Z. trititci* resistance genes *Stb2*, *Stb4*, *Stb7*, and *Stb8* was conducted (Appendix A). For this purpose, the effectiveness of carriers and non-carriers of *Stb2*, *Stb4*, *Stb7*, and *Stb8* genes for the *Z. trititci* score was compared using a non-parametric analog of the analysis of variance and the Kruskal–Wallis test. The wheat samples were grouped into two categories: (1) those with one *Stb* gene and (2) those without the *Stb* gene. In the *Stb2* groups, no significant difference (*p* > 0.05) was observed in either the seedling or adult plant stages (Appendix A). The analysis revealed that there were no statistically significant differences between the carrier and non-carrier groups of other genes (*Stb4*, *Stb7*, and *Stb8*). This is indicated by high *p* values (*p* > 0.05) for both seedling and adult plants (Appendix A). Among the studied genes, the *Stb8* gene demonstrated the greatest influence on resistance to *Z. trititci*, as statistical analysis showed that the lowest *p* value (0.19) was observed between carriers and non-carriers of the Stb8 gene (Appendix A). Hence, the presence of this gene in wheat genotypes significantly enhanced resistance to *Z. trititci*.

However, the significant impact of the year factor on resistance was also observed (*p* < 0.05). Appendix A illustrate the level of *Z. trititci* development in various wheat groups under field conditions across two years (2020 and 2021). The average disease development level in 2020 ranged from 11.4% to 11.8%, whereas in 2021, this figure increased from 15.7% to 16.6% (Appendix A).

To clarify the effects of *Stb* genes on resistance to *Z. tritici*, we compared the field resistance of positive wheat accessions that contained *Stb* genes with the resistance of negative wheat cultivars that lacked *Stb* genes. Statistical parameters were assessed, comparing datasets of a single variable, which consists of two multiple independent levels: (1) *Stb*-positive (27 wheat genotypes carrying *Stb* genes, 162 measurements) and (2) *Stb*-negative (9 genotypes lacking *Stb* genes, 55 measurements). The general variable showed that the population distribution between the groups falls within the following range: Min.—0.0; 1st Qu and Median—10; Mean—18.7; 3rd Qu.—25.0; Max.—65.0 (Appendix A). The average severity of *Z. tritici* in 2020 and 2021 was 32.21% and 37.21%, respectively (Figure 3). The average *Z. tritici* severity among wheat genotypes in the positive group (*Stb* gene carriers) was 16.25%, while in the negative group (lack of *Stb* genes), it was 20.25%. A statistically significant difference was observed between the evaluated levels of the positive and negative genotype groups (*p* value < 0.01). In the positive genotype group, 19% of plants exhibited immunity to the disease (0% severity). The majority of plants in this group (75%) demonstrated resistance to *Z. tritici*. The remaining 25% of plants were susceptible to *Z. tritici*. In the negative genotype group, no wheat varieties exhibited immunity; 29% of plants showed moderate susceptibility, while 71% of plants were susceptible to *Z. tritici* (Appendix A). Thus, a significant influence of genetic background (positive and negative samples concerning the presence of *Stb* genes) on *Z. tritici* severity in the adult plant stage (field studies) was detected (*p* value < 0.01) (Appendix A).

Similar calculations were conducted to obtain data confirming that *Stb* genes influence resistance in the seedling stage. However, no influence of the genetic background factor (positive and negative samples concerning the presence of *Stb* genes) on *Z. tritici* severity in the seedling stage was found (*p* value < 0.9). This suggests that the presence of *Stb* genes does not necessarily imply a causal relationship. Our statistical analyses demonstrated a significant influence of the presence of *Stb* genes on enhancing resistance only in the adult plant stage (under field conditions).

Thus, *Stb* genes have been identified in 27 varieties. Our analyses suggest that the correlation between the presence of *Stb* genes in these 27 varieties and resistance is supported by the provided field resistance data, but not supported by data on seedling resistance.

## 3. Discussion

The main agricultural crop, wheat, is affected by pathogens of leaf spot diseases (LSDs), including Septoria. *Z. tritici* is the main threat to global food security. Losses due to this disease can reach up to 50% in epidemic years and often vary between 5 and 20% depending on the environment and the cultivar of wheat [4]. The species composition of pathogen populations of wheat Septoria spot in Northern Kazakhstan in 2018–2019 comprised three species of Septoria fungi: *Zimoseptoria tritici*, *Septoria nodorum*, and *Septoria avenae* f. sp. *triticea*. Soft spring wheat was mainly affected by *Z. tritici*. During the two-year study of species diversity of Septoria spot pathogens, *Z. tritici* was predominant, followed by *S. nodorum* and *S. avenae* f. sp. *triticea* [5]. This study has significance for future wheat breeding in Kazakhstan. There is not a single variety in our country that is resistant to *Z. tritici* among the varieties approved for use [5]. As a result of our study, identifying the sources of STB genes with minor-to-moderate effects will evolve as an effective way to have durable and resistant wheat varieties for STB.

The cultivation of resistant varieties is the most reliable, environmentally friendly, and cost-effective method of combating diseases. However, traditional breeding methods are not always effective, as the development of resistant varieties is a laborious and lengthy process. In this regard, the use of the marker-assisted selection (MAS) approach accelerates the development of new cultivars. Molecular markers are useful for identifying lines with multiple genes and for pyramiding multiple resistance genes, which is difficult and sometimes impossible to achieve using only phenotypic data [31]. Diversity in *Stb* genes in commercial cultivars could play an important role in managing frequent leaf blotch disease epidemics in the region. In the present study, linked marker analysis revealed the presence of several genes. Five SSR markers, *WMS389*, *WMS111*, *WMS44*, *WMC313*, and *WMS577*, linked with Septoria resistance genes *Stb2*, *Stb4*, *Stb5*, *Stb7*, and *Stb8* were used to confirm these markers in wheat genotypes.

In different previous studies, the sources of *Stb* resistance genes were identified in wheat breeding material [32,33,34,35,36,37,38,39,40]. As a consequence of molecular genetic tagging of resistance genes in 36 varieties of wheat, it was determined that the majority of the analyzed samples carried resistance genes ineffective against *Z. tritici* in their genotype. The result of a study by Babkenova et al. (2017) showed moderately effective resistance of the *Stb2* gene [32]; eight wheat entries with this gene were identified among the studied cultivars. Research conducted by Pakholkova et al., 2016 [33], showed that six of the eight known resistance genes (*Stb1-Stb5*, *Stb7*) had only partial functionality in natural populations of *Z. tritici* in Russia; the *Stb6* gene was highly effective against five populations of *Z. tritici*, and the *Stb8* gene was efficient in an absolute sense against all tested isolates. They also, as in the present study, report the usefulness of screening for the presence of *Stb* genes for the breeding of Septoria-resistant cultivars.

In this study, 60 wheat cultivars and breeding lines were tested with linked markers for some *Stb* genes. The proportion of wheat entries with *Stb* genes was quite high at twenty-seven of the genotypes tested (45.0%), including four from Kazakhstan, nine from Russia, nine from the CIMMYT-ICARDA-IWWIP program, and five from the CIMMYT-SEPTMON nursery. Molecular screening of these genotypes showed contrasting differences in the gene’s frequencies. Among the 60 entries, 10 (16.7%) carried resistance genes *Stb2* and *Stb8.* Gene *Stb4* was found in seven cultivars (6.7%), while *Stb5* was not detected in any genotypes in this study. Omskaya 18 (*Stb2* and *Stb7*), EFED/22150 and Egemen (*Stb2* and *Stb8*), *and* EFED/F5.83 7792(BAJAS) (*Stb4* and *Stb7*) had the highest number of resistance genes, with two-gene combinations. Genotypes Omskaya 18, EFED/22150, EFED/F5.83 7792(BAJAS), and Egemen were identified with a maximum of two *Stb* genes, followed by one *Stb* gene in twenty-four genotypes. The used SSR markers did not identify *Stb* genes in 32 genotypes. None of the five *Stb* genes were detected in a few resistant and moderately resistant genotypes (JAC161/TEMU51.80, NANJTNG82149/KAUZ, KR11-9025, KR12-07, P83-5112/V82274, Sapaly, and BR14/CEP847-2), suggesting that additional *Stb* genes confer resistance to Septoria in these genotypes. This demonstrates the diversity of *Stb* genes in the gene pool comprising cultivars and advanced breeding lines of wheat in the studied collection.

The wheat genotypes from Kazakhstan, Russia, and CIMMYT differed greatly in Septoria severity recorded in the adult plant stage in the field in Kazakhstan. This supports reports on varietal resistance and variation among *Z. tritici* populations in Kazakhstan and Russia [4,33,41,42,43,44,45,46]. Zeleneva and Konkova (2023a) evaluated the resistance of winter and spring wheat cultivars to detect resistance to *Septoria blotch* and studied the populations of *Parastagonospora nodorum* and *P. pseudonodorum* in the territory of the Saratov region of Russia to detect the presence of effector genes; eleven wheat genotypes were selected as resistant to different species of Septoria blotch (*Zymoseptoria tritici*, *P. nodorum*, *P. pseudonodorum*) [41].

In our study, of greatest interest are six wheat genotypes—carriers of effective *Stb* genes, including three entries from CIMMYT and three entries from Russia, which demonstrated a combination of field and seedling resistance. Among them, three single cultivars (KR11-03, Omskaya 35, and KR11-9014) separately carried the *Stb2* gene; two cultivars (CROC_1/AE.SQUARROSA-205//BORL95/3/2/*Milan и Saratovskaya 55) carried the *Stb8* gene, and one cultivar (Omskaya 29) carried the *Stb4* gene.

It was found that the year significantly influences the development of *Z. trititci* in field conditions between the carrier and non-carrier groups of *Stb2*, *Stb4*, *Stb7*, and *Stb8* genes. This indicates that the influence of genetic factors may vary depending on the conditions of the year. The average level of *Z. trititci* in 2021 was higher than in 2020 for both carriers and non-carriers of *Stb* resistance genes. This may suggest an increase in the infection rate in the latter year, regardless of the presence or absence of specific genetic variants.

The result of a recent study on screening the resistance of spring bread wheat cultivars and lines from Russia to Septoria leaf blotch allowed us to identify 23 accessions with high levels of resistance to *Z. tritici* [47]. In our study, twenty-two genotypes (Severyanka, Rosinka 3, Omskaya 18, Omskaya 28, Omskaya 29, Omskaya 35, Omskaya 36, Mironovskaya 808, Pamyati Azieva, KR11-40, KR11-03, NANJTNG82149/KAUZ, CROC 1AE. SQUARROSA, GAN/AE.437. SQUARROSA, TRAP#1/BOW, EFED/F5.83 7792(BAJAS), Batyr, Bayandy, Samgay, and Sapaly) showed ≤10% disease severity in both years. Resistant reactions to *Z. tritici* isolates in the seedling stage were associated with adult plant resistance to disease in four wheat entries, including Rosinka 3, Omskaya 18, Omskaya 29, and KR11-03. These genotypes were highly resistant under both greenhouse and field conditions. The resistance of Rosinka 3 was due to the presence of *Stb8.* Omskaya 18 showed an immune reaction in the field and an MS reaction to *Z. tritici* in the seedling stage, possibly provided by the combination of *Stb7* and *Stb2* genes. The high resistance in both the adult and seedling stages of Omskaya 29 and KR11-03 was due to *Stb4* and *Stb2* genes and, possibly, by the presence of unknown genes. The deployment of specific gene combinations provides durable and improved resistance versus using single genes because a single specific gene is subject to becoming susceptible due to genetic shifts in the pathogen [48]. Given that the assessed entries comprised germplasm from Russia and CIMMYT, developed within breeding programs aimed at enhancing resistance to Septoria, it is probable that they possess diverse constitutions of resistance genes. Among these sources, twenty-two entries showed disease resistance, suggesting their potential value as sources of resistance, and can be used directly in breeding programs to improve the Septoria resistance of wheat. Marker-assisted selection can be efficiently applied to develop wheat cultivars with effective *Stb* gene combinations that would directly assist in developing durable resistance in Kazakhstan. Resistant genotypes could also be used as improved parents in cross-breeding programs to develop new varieties.

## 4. Materials and Methods

### 4.1. Plant Material

This study used 60 winter wheat genotypes comprising both cultivars and breeding lines. The collection included 15 cultivars and breeding lines from Kazakhstan, 15 cultivars from Russia, 15 lines released by the CIMMYT-ICARDA-IWWIP program, and 15 lines released by the CIMMYT-SEPTMON nursery (Table 3). The collection comprised several important wheat genotypes that have been widely used as parents in breeding programs across Kazakhstan and Central Asian countries. The highly susceptible control cultivar Morocco as well as the isogenic lines with known resistance genes (Veranopolis (*Stb2*), Tadinia (*Stb4*), CS (Synthetic 7D) (*Stb5*), Estanzuela Federal (*Stb7*), and Synthetic W7984 (*Stb8*)) were included as checks.

### 4.2. Field Evaluation of Wheat Entries against Zymoseptoria tritici

Evaluation of field resistance to *Septoria tritici* was carried out at the Kazakh Research Institute of Agriculture and Crop Production (KazNIIZiR), (Almalybak, 43°13′09″ N, 76°36′17″ E, Almaty region) in Southeast Kazakhstan, Almaty region, during the 2020 and 2021 cropping seasons.

Each entry was planted in a 1 m^2^ plot in mid-September and was harvested in mid-August the following year for two years. Experiments were conducted with a completely randomized design with two replicates in 1 m^2^. The *Septoria*-susceptible cultivar Morocco was planted in every 10th row and as a spreader border around the nursery to ensure uniform infection. Fertilizer treatments of 60 and 30 kg/ha of N and P_2_O_5_, respectively, and other management practices corresponded to those normally recommended for the region [49]. Annual rainfall ranged from 332 to 644 mm during the two years. The growing seasons were favorable for pathogen infection and disease development. The mean daily temperature and relative humidity showed similar trends in both years and conditions were highly conducive for *Septoria* infection and development.

The experiments on an artificial infectious background were made with naturally infected straw stubbles. Ten Flag-1 leaves were evaluated for each disease assessment of genotypes. The disease was assessed three times. In October, before sowing, the infected straw (1 kg/m^2^) was incorporated into the soil. For the evaluation of field response, disease severities were assessed on first leaves and flag leaves when all wheat genotypes were near or at Zadoks growth stages Z69 (flowering) and Z75 (milk) [50]. The percentage of *Septoria*-infected leaf area was determined on each leaf and the average value for all evaluated leaves was calculated for each wheat entry to determine the STB score. A rating system based on the % leaf area infected, developed for appraising the foliar intensity of diseases, was used to categorize host reactions to *Septoria* according to a double-digit scale (00–99) modified from Saari and Prescott [51]. According to the degree of damage, the varieties were divided into the following groups: 0–10%—highly resistant (RR; free from infection or with a few isolated lesions on the lowest leaves only); 11–20%—resistant (R; scattered lesions on the second set of leaves with first leaves infected at light intensity); 21–40%—moderately susceptible (MS; moderate-to-severe infection of lower leaves with scattered-to-light infection extending to the leaf immediately below the mid-point of the plant); 41–70%—susceptible (S; severe lesions on lower and middle leaves; moderate to severe infection of upper third of plant; flag leaf infected in amounts more than a trace), 71–100%—highly susceptible (SS; severe infection on all leaves and the spike infected to some degree).

### 4.3. Inoculum Production and Inoculations and Seedling Resistance

The plant material of wheat was inoculated with a mix of *Zymoseptoria tritici* isolates. The isolates used in this study were previously obtained from bread wheat (*Triticum aestivum* L.) and durum wheat (*Triticum durum* Desf) [52]. The isolates *Z. tritici*, 156-22-*Z. tritici*, 154-22-*Z. tritici*, 1-22-*Z. tritici*, 6-22-*Z. tritici*, 3-22-*Z. tritici*, were collected in 2022 from the spring durum wheat cultivar Valentina (Triticum durum Desf.) in the Saratov region (51°34′28″ N, 46°00′24″ E), and from winter bread wheat cultivars Al’mera and Astet in the Tambov region (52°40′25″ N, 41°10′20″ E) of Russia. These five isolates of *Z. tritici* were used for inoculum production in this study. Stock cultures were cultivated on yeast sucrose agar (YSA; 10 g L^−1^ yeast extract, 10 g L^−1^ sucrose, 1.2% agar) supplemented with kanamycin (50 µg/mL) [53].

To test the virulence of these 5 isolates, wheat cultivars (durum wheat: Valentina cv; bread wheat: Al’mera and Astet cvs) were inoculated with *Z. tritici* isolates in a greenhouse environment. After 4 days, leaf segments from seedlings were cut and put on agar in Petri dishes at 20 °C for subsequent culture and monoconidial isolation, identification, and storage. Three weeks later, a high presence of pycnidia was observed. The *Z. tritici* isolates were pathogenic to both wheat species. The culture of *Z. tritici* was revitalized by transferring it to fresh PDA medium (potato dextrose agar, 39 g/L HiMedia, Mumbai, India) and incubated at 18 °C with a photoperiod of 12 h over 10 days. After incubation, sterile distilled water was added to each plate and spores were scraped gently with special glass rods. The spores were then transferred to a yeast–sucrose liquid medium (yeast extract 10 g/L; sucrose 10 g/L) and left shaking for 7 days at 18 °C, with permanent light. The spores were collected by centrifugation at 5000 rpm for 5 min at 15 °C, washed twice with sterile distilled water, and resuspended in a MgSO_4_ solution (0.01 M) containing Tween 20 surfactant (0.1% *v*/*v*). The concentration was adjusted to 10^7^ spores/mL [52]. The dishes inoculated with *Z. tritici pycnidia* were placed in a thermostat at a temperature of 21 °C; no lighting was required. All Petri dishes were sealed with a double layer of paraffin. The cultures were monitored daily. On days 8–10, young colonies were transferred to a fresh nutrient medium. The fungi were cultivated for 30 days in a thermostat at a temperature of 21 °C. For inoculation, 30-day-old colonies were used or they were stored in a refrigerator at a temperature of +4 °C [53].

Soil preparation and inoculation were carried out according to generally accepted methods. The universal substrate was used (“Terra vita”, manufactured by OOO “Nord Pflp”, a limited liability company under the laws of the Russian Federation). Ten seeds of all 60 wheat accessions were raised in plastic containers with a capacity of 20 cm^3^. Ten seeds of each wheat variety were sown in a pot. All entries were arranged in a randomized complete design with three replications. So, 10 seedlings in a pot and 30 plants of the same variety were evaluated. Each container was considered as an experimental unit, and every single plant of a wheat variety in a container with three lots of ten seedlings served as an entry. The fungal isolates were mixed before inoculation. Thus, ten seedlings of each genotype were inoculated individually at the two-leaf stage against the mix of *Z. tritici* isolates. In each experiment, replications were treated as random effects, and the wheat accessions as fixed effects. All experiments were conducted in a greenhouse facility at the All-Russian Institute of Plant Protection (ARIPP, St Petersburg, Russia). The cultivar Morocco, susceptible to *Z. tritici*, was included as a check to monitor the development of infection. Spores produced by fungal colonies in pure culture were used as the inoculum. Inoculation was carried out with an aqueous suspension of spores. One day before inoculation, the viability of the existing inoculum was determined; then, a working suspension of spores was prepared [54]. Cells were scraped from the colony surface using a sterile pipet tip and resuspended in a sterile 2 mL microcentrifuge tube containing 1 mL of sterile water. Dilutions (1:10) of the blastospore suspension were made in additional sterile 2 mL microcentrifuge tubes. Tubes were briefly vortexed between dilutions.

The hemocytometer and coverslip were carefully cleaned with 70% ethanol before use and the coverslip was positioned correctly (indicated by the appearance of Newton’s rings). An appropriate dilution was selected; the blastospore suspension was briefly vortexed, and 10 µL was pipetted into each compartment of the hemocytometer. A compound microscope was used to observe cells. Cell density was calculated considering the average number of counted cells, the specifications of the hemocytometer in use, and the dilution of the cell suspension that was counted [54]. The inoculum was adjusted to the required cell density (e.g., 10^6^ or 10^7^ cells/mL) in 0.1% Tween^®^ 20 in sterile water. The final inoculum was kept in a sterile 50 mL Falcon tube and used within a few hours. The suspension was evenly applied to the plants using a spray bottle, after which the cuvettes with flowerpots were placed in a climate chamber (model MLR-352H-PE, “PHCbi”, Tokyo, Japan).

To inoculate the plants for each treatment, a cleaned spray gun was used with a pressure of 2.0 bar to inoculate the marked leaf sections of each plant with the *Z. tritici* inoculum by evenly spraying until runoff. The inoculum was left to dry on the leaf surface (~15 min) and the pots were placed (sorted by treatment) in large plastic bags containing water (~1 L). The bags were sealed using tape or plastic locking clips to generate an environment with maximal relative humidity. The plants were incubated in the bags at ~20 °C (day)/~12 °C (night) and a 16 h day/8 h night cycle under controlled greenhouse conditions. After 48 h, the pots were removed from the bags, and the plants were placed back onto the trays. The randomized placement was ensured. The plants were watered regularly and a relative humidity of 70% to 90% was maintained. The wheat was grown for 21 days post-inoculation (dpi) in the same temperature and light conditions [54,55]. The degree of plant damage was determined 20–22 days after inoculation. Seedlings were rated using the rating scale of 0–4 [43], where 0–1 is resistant, 2 is moderately susceptible, 3 is susceptible, and 4 is highly susceptible. According to the methodological recommendations [55], we evaluated 10 seedlings separately in each pot. Since we had 3 replicates, we received 30 ratings per variety. Then, the results were statistically processed.

### 4.4. DNA Extraction and Molecular Screening of Stb Resistance Genes

Each genotype’s genomic DNA was extracted using the CTAB method from fresh leaves of individual plants at the two-leaf seedling stage [56]. The concentration and purity of the resulting preparation were measured using a NanoDrop One spectrophotometer. The DNA concentration for PCR was adjusted to 20 ng/µL. Primers linked to *Stb* genes were employed according to certain approved protocols. The polymerase chain reaction (PCR) was conducted using the primers and annealing temperature settings that were specified for each *Stb* gene in the references (Table 4). A Bio-Rad T100TM Thermal Cycler (Bio-RAD, Hercules, CA, USA) was used to conduct the PCR experiments. The PCR mixture (25 µL) contained 2.5 µL of genomic DNA (30 ng), 1 µL of each primer (1 pM/µL) (Sigma-Aldrich, St. Louis, MO, USA), 2.5 µL of dNTP mixture (2.5 mM, dCTP, dGTP, dTTP and dATP aqueous solution) (ZAO Sileks, Moscow, Russia), 2.5 µL of MgCl2 (25 mM), 0.2 µL of Taq polymerase (5 units µL) (ZAO Sileks, Russia), 2.5 µL of 10X PCR buffer, and 12.8 µL of ddH20. TBE buffer (45 mM Tris-borate, 1 mM EDTA, pH 8) was used to separate the amplification products, and ethidium bromide was added [57]. A 100 bp DNA ladder (Fermentas, Vilnius, Lithuania) was employed to gauge the size of the amplification fragment. The Gel Documentation System (Gel Doc XR+, BIO-RAD, Hercules, CA, USA) was used to visualize the results. Each sample underwent three separate tests.

Leaf samples from all 60 entries were genotyped with the SSR marker designed to detect alleles of the *Stb* genes. Identification of *Stb* genes resistant to the *Z. tritici* pathogen was carried out using molecular genetic tagging in selected winter wheat cultivars. To analyze *Stb* genes of resistance, microsatellite markers linked to the resistance genes were used (Table 3) [20,30,58,59,60]. The isogenic lines with known resistance genes were used as control entries. To investigate the presence of the *Stb2* resistance gene, the SSR marker WMS389 was used with the Veranopolis wheat cultivar as a positive control. The presence of Stb4 was studied using the SSR marker *WMS111* and Tadinia as positive control. The presence of *Stb5* was confirmed using the *WMS44* marker when the *Stb5*-carrying cultivar CS (Synthetic 7D) was used as a positive control. The *Stb7* gene was identified using SSR marker *WMC313* and cultivar Estanzuela Federal as a positive control, and the presence of the *Stb8* gene was confirmed using the *WMS577* marker when the *Stb8*-carrying Synthetic W7984 was used as the positive control.

### 4.5. Statistical Data Processing

The following formula, developed by Wilcoxson et al. [61], was used to determine the area under the disease progress curve (*AUDPC*):(1)$$AUDPC=\sum_{i=1}^{n-1}\frac{y_{i}+y_{i+1}}{2}\times \left(t_{i+1}-t_{i}\right)$$

*y_i_*—an evaluation of disease at the *i*th observation;

*t*_i_—time (in days) at the *i*th observation;

*n*—the total number of observations.

To determine genotypic and year variances among genotypes for traits of *Zymoseptoria tritici* resistance, an analysis of variance (ANOVA) was performed using the R-Studio, R 4.3.3 version software according to the nonparametric Wilcoxson and Kruskal–Wallis tests. The significance of the calculations was assessed using the *p*-value [62], and coefficients of Pearson correlation were calculated using the mean values of the characters assessed [63]. The broad-sense heritability index, which measures the percentage of phenotypic variation attributable to genetic determinants, was derived using the ANOVA results: hb^2^ = SSg/SSt, where SSg is the sum of squares for a genotype and SSt is the total sum of squares.

## Figures and Tables

**Figure 1 plants-13-01113-f001:**
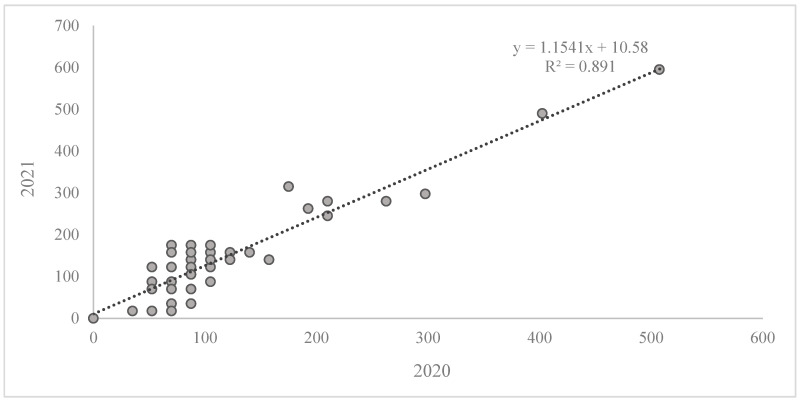
The resistance score for *Z. tritici* with the AUDPC values for 2020 and 2021 for wheat genotypes in the adult plant stage.

**Figure 2 plants-13-01113-f002:**
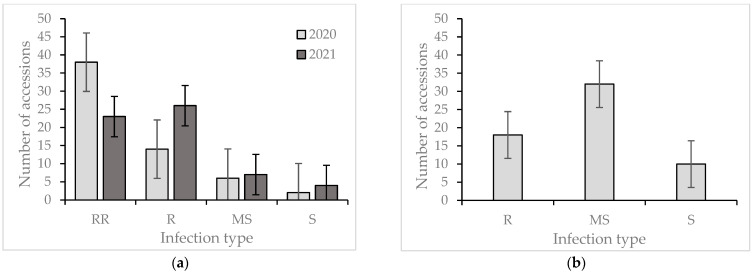
Summary of infection types among 60 wheat cultivars and breeding lines infected with *Z. tritici* (**a**) in the adult plant stage and (**b**) in the seedling stage. Note: RR—highly resistant; R—resistant; MS—moderately susceptible; S—susceptible.

**Figure 3 plants-13-01113-f003:**
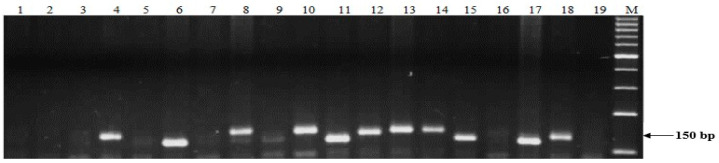
DNA amplification products for wheat cultivars and breeding lines using primers to the SSR *WMS389* locus linked to the *Stb2* resistance gene. The arrows show the band size of the *Stb2*-carrying germplasm (150 bp). The sizes of the bands for *Stb2* are 100 bp (lanes 5, 6, 11, 15, 17, and 18—positive control). Fragments amplified by *WMS389* were separated in the 2% agarose gels. Lane: 1—Saratovskaya 70, 2—KR12-5075, 3—Batyr, 4—Kazakhstanskaya 4, 5—Omskaya 19, 6—SOMO/SORA/ACTS5, 7—Severyanka, 8—KR12-5001, 9—Kyzylbiday, 10—TOO11/TOOOO7, 11—KR11-03, 12—TALHUENJNJA, 13—Botagoz, 14—DOMOJNJA, 15—Omskaya 38, 16—F133/SHA5//OPATA, 17—Sultan 2, 18—*Stb2* (positive control), 19—Morocco (negative control); M -molecular weight marker (Gene-Ruler, 100 bp DNA ladder).

**Table 1 plants-13-01113-t001:** Analysis of variance of the effect of the plant genotype and pathogen on the resistance of wheat seedlings to *Zymoseptoria tritici*.

Trait	Factor	SS	df	MS	F-Value	%SS	h_b_^2^, %
AUDPC, field	Genotype	1,197,935	59	20,304	25.72 ***	94.44	0.94
	Year	24,013	1	24,013	30.42 ***	1.89	
	Residuals	46,578	59	46,578		3.67	
*Z. tritici*, seedling	Genotype	134.87	59	2.2859	11.82 ***	74.24	0.74
	Replication	1.17	4	0.2917	1.50	0.64	
	Residuals	45.63	236	0.1934		25.12	

Notes: SS—a sum of squares; df—degree of freedom; MS—mean squares; h_b_^2^—broad-sense heritability index. *** Significant difference at *p* < 0.001.

**Table 2 plants-13-01113-t002:** Genetic parameters of wheat genotypes.

Trait	Min	Max	Mean	SE	Variance	SD	Median	Mode	CV
AUDPC 2020	0	507.5	114.92	11.87	8454.23	91.95	87.5	70	80.01
AUDPC 2021	0	595	143.21	14.51	12,639.21	112.42	122.5	70	78.50
*Z. tritici*, seedling	1	3.6	2.23	0.09	0.457175	0.68	2.4	2.6	30.28

Notes: SD—significant difference; CV—coefficient of variation.

**Table 3 plants-13-01113-t003:** Disease severity for *Z. tritici* and detected *Stb* genes based on linked markers in the collection of wheat genotypes.

Name of Variety	Origin	Type	Final Score 2020 (Field)	AUDPC 2020 (Field)	Final Score 2021 (Field)	AUDPC 2021 (Field)	*Z. tritici* (Seedling)	*Stb* Genes
Saratovskaya 70	RU	Spring	15	175	40	315.0	2.6	none
Severyanka	RU	Spring	5	52.5	10	70.0	2	none
Saratovskaya 29	RU	Spring	15	105	20	157.5	2	none
Saratovskaya 42	RU	Spring	15	87.5	25	175.0	1.2	none
Saratovskaya 55	RU	Spring	10	70	20	175.0	1.6	*Stb8*
Albidum 31	RU	Spring	5	70	10	70.0	3.6	*Stb4*
Rosinka 3	RU	Spring	10	87.5	10	70.0	2.6	*Stb8*
Omskaya 18	RU	Spring	0	0	0	0.0	2.4	*Stb7*, *Stb2*
Omskaya 19	RU	Spring	10	87.5	20	157.5	2.2	none
Omskaya 28	RU	Spring	5	35	5	17.5	2.6	none
Omskaya 29	RU	Spring	5	70	10	70.0	1.4	*Stb4*
Omskaya 35	RU	Spring	5	52.5	15	87.5	1.4	*Stb2*
Omskaya 36	RU	Spring	5	70	10	70.0	2.6	*Stb7*
Mironovskaya 808	RU	Winter	10	105	10	122.5	3	*Stb8*
Pamyati Azieva	RU	Spring	5	52.5	10	70.0	2.6	*Stb2*
KR11-13	CIMMYT-ICARDA-IWWIP	Winter	10	122.5	15	140.0	2.4	*Stb4*
KR12-9011	CIMMYT-ICARDA-IWWIP	Winter	15	105	20	157.5	3	*Stb8*
KR11-40	CIMMYT-ICARDA-IWWIP	Winter	10	87.5	10	140.0	2	none
KR12-9012	CIMMYT-ICARDA-IWWIP	Winter	15	87.5	20	157.5	3	*Stb7*
KR11-9025	CIMMYT-ICARDA-IWWIP	Winter	15	105	10	122.5	1.4	none
KR12-5001	CIMMYT-ICARDA-IWWIP	Winter	5	52.5	15	122.5	2.4	none
KR12-07	CIMMYT-ICARDA-IWWIP	Winter	20	157.5	15	140.0	1.4	none
KR12-5035	CIMMYT-ICARDA-IWWIP	Winter	10	70	15	122.5	2.8	none
KR12-09	CIMMYT-ICARDA-IWWIP	Winter	5	70	10	70.0	2.2	none
KR12-5075	CIMMYT-ICARDA-IWWIP	Winter	30	210	30	245.0	3	*Stb8*
KR11-03	CIMMYT-ICARDA-IWWIP	Winter	10	87.5	10	122.5	1	*Stb2*
KR12-10	CIMMYT-ICARDA-IWWIP	Winter	10	122.5	15	140.0	2.4	*Stb2*
KR11-9014	CIMMYT-ICARDA-IWWIP	Winter	30	297.5	25	297.5	1	*Stb2*
KR12-18	CIMMYT-ICARDA-IWWIP	Winter	5	105	15	140.0	2	*Stb4*
KR11-26	CIMMYT-ICARDA-IWWIP	Winter	20	192.5	30	262.5	2.6	*Stb4*
SOMO/SORA/ACTS5	CIMMYT—SEPTMON	Spring	10	70	20	122.5	3	*Stb2*
P83-5112/V82274	CIMMYT—SEPTMON	Spring	15	140	20	157.5	1.6	none
DOMOJNJA	CIMMYT—SEPTMON	Spring	10	70	15	87,5	2	none
NANJTNG 82149 KAUZ	CIMMYT—SEPTMON	Spring	10	87.5	20	122.5	1.2	none
CROC_1/AE.SQUARROSA-205//BORL95/3/2/*Milan	CIMMYT—SEPTMON	Spring	5	52.5	10	70.0	1.6	*Stb8*
TALHUENJNJA	CIMMYT—SEPTMON	Spring	15	122.5	20	157.5	2.4	none
JAC161/TEMU51.80	CIMMYT—SEPTMON	Spring	5	70	20	157.5	1	none
GAN/AE.437 SQUARROSA	CIMMYT—SEPTMON	Spring	5	70	10	70.0	2.4	none
EFED/22150	CIMMYT—SEPTMON	Spring	15	105	15	87.5	2.6	*Stb2*, *Stb8*
BR14/CEP847-2	CIMMYT—SEPTMON	Spring	15	87.5	20	105.0	1.6	none
ECHA/LI115	CIMMYT—SEPTMON	Spring	10	105	15	140.0	1.8	none
TOO11/TOOOO7	CIMMYT—SEPTMON	Spring	10	52.5	15	87.5	2.8	none
F133/SHA5//OPATA	CIMMYT—SEPTMON	Spring	10	70	15	87.5	2.6	none
TRAP#1/BOW	CIMMYT—SEPTMON	Spring	5	52.5	10	70.0	2.4	*Stb4*
EFED/F5.83 7792(BAJAS)	CIMMYT—SEPTMON	Spring	5	52.5	10	70.0	2.4	*Stb7*, *Stb4*
Batyr	KZ	Winter	10	70	10	35.0	2.8	none
Bayandy	KZ	Winter	0	52.5	5	17.5	2.6	none
Botagoz	KZ	Winter	10	87.5	10	35.0	3	none
Celinnaya 50	KZ	Spring	15	105	25	175.0	1.6	none
Derbes	KZ	Winter	10	157.5	15	140.0	2.8	none
Egemen	KZ	Winter	45	507.5	50	595.0	2.6	*Stb2*, *Stb8*
Kazakhstanskaya 4	KZ	Spring	25	210	30	280.0	3.6	none
Kokbiday	KZ	Winter	5	70	5	17.5	3	*Stb8*
Kupava/Avocet (S)-1	KZ	Winter	15	87.5	20	157.5	2.6	*Stb8*
Kyzylbiday	KZ	Winter	30	402.5	40	490.0	2.4	none
Manshyk	KZ	Winter	40	402.5	40	490.0	1	none
Samgay	KZ	Spring	5	70	10	70.0	2.6	none
Sapaly	KZ	Winter	10	87.5	10	70.0	1.6	none
Steklovidnaya 24	KZ	Winter	35	262.5	30	280.0	1	none
Sultan 2	KZ	Winter	10	122.5	15	140.0	3	*Stb2*
Controls								
Morocco	MA	Winter	50	595	60	665	3.8	none
Veranopolis (*Stb2*)	Brazil	Spring	5	52.5	10	70	1	*Stb2*
Tadinia (*Stb4*)	USA	Spring	5	70	10	70.0	1.6	*Stb4*
CS (Synthetic 7D) (*Stb5*)	USA	Spring	5	52.5	10	70.0	1.4	*Stb5*
Estanzuela Federal (*Stb7*)	Uruguay	Spring	5	52.5	10	70.0	1	*Stb7*
Synthetic W-7984 (*Stb8*)	USA	Winter	5	52.5	10	70.0	1.2	*Stb8*

Notes: According to the degree of damage, the varieties were divided into the following groups: 0–10%—highly resistant (RR); 11–20%—resistant (R); 21–40%—moderately susceptible (MS); 41–100%—susceptible (S). Final score—latest evaluation of plant resistance in the field. AUDPC—area under the disease progress curve. Seedlings were rated using the rating scale 0–4 [29,30]: 0–1—resistant, 2—moderately susceptible, 3—susceptible, and 4—highly susceptible. *Z. tritici* (seedling)—reaction to infection with a mix of pathotypes of *Z. tritici* in the seedling stage. *Stb* genes—detected *Stb* genes based on linked markers in the collection of wheat genotypes.

**Table 4 plants-13-01113-t004:** Molecular markers used to identify *Stb* genes.

Gen	Chr	Type of Marker	Primer Name	Sequence of Primers 5′-3′	Annealing Temperature, °C	Fragment Size, b.p	Reference
Stb2	3BS	SSR	WMS389-L	5′-ATC ATG TCG ATC TCC TTG ACG-3′	60	150	[58]
			WMS389-R	5′-CAT GCA CAT TTA GCA GAT-3′			
Stb4	6D	SSR	WMS111-L	5′-ACC TGA TCA GAT CCC ACT CG-3′	55	210/220	[30]
			WMS111-R	5′-TTC GTA GGC TCT CTC CGA CTG-3′			
Stb5	7DS	SSR	WMS44-L	5′-GTT GAG CTT TTC AGT TCG GC-3′	60	182	[59]
			WMS44-R	5′-ACT GGC ATC CAC TGA AGC TG-3′			
Stb7	4AL	SSR	WMC313-L	5′-GCA GTC TAA TTA TCT GCT GG CG-3′ 5′-GGG TCC TTG TCT ACT CAT GT CT-3′	51	197	[60]
			WMC313-R				
Stb8	7BL	SSR	WMS577-L	5’-ATG GCA TAA TTT GGT GAA AT TG-3 5′-TGT TTC AAG CCC AAC TTC TA TT-3′	55	180	[20]
			WMS577-R				

## Data Availability

The data presented in this study are available on request from the corresponding author. The data is publicly available.

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
