# Peer review of "Identification of Wheat Septoria tritici Resistance Genes in Wheat Germplasm Using Molecular Markers"

_plants, 2024, doi:10.3390/plants13081113_

Round 1

Reviewer 1 Report

Comments and Suggestions for Authors

Dear Colleagues, 

Congratulations on your good paper. I made a few suggestions for your consideration. 

Good luck,

Hafiz 

Comments on the Quality of English Language

The English could be improved. 

Author Response

Question 1. Does the introduction provide sufficient background and include all relevant references? Are the conclusions supported by the results? It can be improved.

Answer 1. Yes, we improved the introduction.        Now, in the section Introduction we provide detailed information on the current situation in Kazakhstan.

Question 2. Are all the cited references relevant to the research? Must be improved.

Answer 2. Yes, we improved the section References. We added some relevant references.

Question 3. Are the conclusions supported by the results? Can be improved.

Answer 3. Yes, we improved the section Conclusions.

Question 4. The English could be improved.

Answer 4. Yes, we improved the English of the manuscript.

Reviewer 2 Report

Comments and Suggestions for Authors

This can be a useful information for wheat farms.  However, there are many technical information lacking in this manuscript.

1. Lines 111-114 and Table 1.  Please, specify what is the input data for ANOVA.  Otherwise, it is impossible to see what actually goes into your analysis. 

2. Lines 136-141 and Figure 1. Please, add the infection type in the legend.  What is the criteria for RR/R/MS/S as the degree of damage?

3. Lines 195-199: Is there any evidence to support changes in Stb4 genes?

Comments on the Quality of English Language

I suggest minor grammar check.

Author Response

REVIEWER 2 – Answers in MS IN GREEN COLOR

Comments and Suggestions for Authors  

This can be a useful information for wheat farms.  However, there are many technical information lacking in this manuscript.

  1. 1. Lines 111-114 and Table 1. Please, specify what is the input data for ANOVA. Otherwise, it is impossible to see what actually goes into your analysis.

We agree with your comment. For clarification, we have added information about the input data for analysis (lines 112-116).

Analysis of variance (ANOVA) revealed highly significant effects for wheat genotypes on resistance to Zymoseptoria tritici both in field and laboratory tests (Table 1).  A total of 60 cultivars and breeding lines were analyzed in 2 years for field trials and in 3 replicates for laboratory screening of seedling resistance (ASR). The resistance score (AUDPC, field / Z.tritici, seedling) to Z. tritici in ANOVA was taken as a variable; year of field testing (n=2,  for APR) or replication (n=5, for ASR) and genotype (n=60) as a factor. There was significant variation among genotypes and disease severity (p<0.001) in both growing seasons of field trials (p<0.001).

Всего было проанализировано 60 сортов и селекционных линий в течение 2 лет для полевых испытаний и в 3 повторностях для лабораторного скрининга устойчивости проростков. Показатель устойчивости (AUDPC, поле / Z.tritici, рассада) к Z. tritici в ANOVA принимался как переменная, год полевых испытаний (n=2, для APR) или повторность (n=5, для ASR) и генотип (n =60) как фактор.

  1. Lines 136-141 and Figure 1. Please, add the infection type in the legend. What is the criteria for RR/R/MS/S as the degree of damage?

We agree with your comment. Thanks for pointing out the flaw. We have added clarification to the infectious type estimates and updated the figures. (lines 139-140)

                                                            (a)                                                                          (b)

Figure 1. Summary of infection types among 60 wheat cultivars and breeding lines infected with Z.tritici (a) at the adult plant stage and (b) at the seedling stage. Note: RR – highly resistant; R – resistant; MS – moderately susceptible; S – susceptible.

We have added explanations of the criteria for assessing resistance by infectious type in the materials and methods section (lines 339-343)

According to the degree of damage, the varieties were divided into the following groups: 0-10% – highly resistant (RR; Free of infection or few isolated lesions on lowest most leaves only); 11-20% – resistant (R; Scattered lesions on the second set of leaves with first leaves infected at light intensity); 21-40% – moderately susceptible (MS; Moderate to severe infection of lower leaves with scattered to light infection extending to the leaf immediately below the mid-point of the plant); 41-70% – susceptible (S; Lesions severe on lower and middle leaves; moderate to severe infection of upper third of plant; flag leaf infected in amounts more than a trace), 71-100% - highly susceptible (SS; Severe infection on all leaves and the spike infected to some degree).

  1. Lines 195-199: Is there any evidence to support changes in Stb4 genes?

Строки 195-199: Есть ли какие-либо доказательства, подтверждающие вариации в генах Stb4?

There is no information confirming variations in the Stb4 gene. It is known that the Stb4 locus from the wheat cv. Tadinia showed resistance to Z. tritici at both seedling and adult-plant stages [46]. Similar results were obtained in our studies. Based on the molecular screening results, all of the seven genotypes containing Stb4 genes showed fairly good to moderate resistance responses at field testing. The Stb4 gene had a large effect on Z. tritici, which provided a high level of field resistance in both years with a final disease score of 5-10%. These genotypes, as a role, exhibited also seedling resistance values (from moderately susceptible to susceptible).

Line 204: In answer to your question, I discovered an error in calculating the percentage of carriers of the Stb4 gene. Out of 60 samples, the gene was found in 7, which is not 20%, but 11.6%. The error has been fixed.

Answer Line 204: Of the 60 cultivars/lines identified to carry this resistance gene in our study, 7 genotypes (11,6%) amplified 210/220-bp fragment indicating the presence of the Stb4 resistance gene. These wheat entries include Albidum 31, Omskaya 29, KR11-13, KR12-18, KR11-26, TRAP#1/BOW and EFED/F5.83-7792(BAJAS) (Table 3)

Reviewer 3 Report

Comments and Suggestions for Authors

Wheat Septoria tritici disease caused by Zymoseptoria tritici (Z. tritici) is the main threat to global food security: one of the most serious threats to wheat crops, causing severe yield losses worldwide, including in Kazakhstan. Twenty-one qualitatively inherited major genes (Stb1 to Stb18, StbSm3, StbWW, and TmStb1) have been mapped on wheat chromosomes against the STB pathogen. The objectives of this study were to: (1) evaluate a wheat collection for Z. tritici resistance during an adult plant and seedling growth stages, (2) identify the sources of resistance genes that provide resistance to Z. tritici using molecular markers linked to Stb genes, (3) identify potentially useful resistant wheat genotypes among cultivars and advanced breeding lines.

The authors evaluated 60 winter and spring wheat genotypes for Z. tritici resistance. According to the field reactions, 22 entries (35,7%) showed ≤10% disease severity. For example, the resistance of Rosinka 3 was due to the presence of Stb8, Omskaya 18 showed an immune reaction in the field and a moderately susceptible reaction at the seedling stage possibly provided by a combination of Stb7 and Stb2 genes. The high resistance both at adult and seedling stages of Omskaya 29 and KR11-03 was due to Stb4 and Stb2 genes and, possibly, by the presence of unknown genes. Among the 60 entries, 10 (16.7%) carried resistance genes Stb2 and 29 Stb8, and the gene Stb4 was found in seven cultivars (6.7%). Marker-assisted selection can be efficiently applied to develop wheat cultivars with effective Stb gene combinations that would directly assist in developing durable resistance in Kazakhstan.

The objectives of this study are clear and the experiments are well done. The manuscript needs some improvement.

1.       Figure 1 caption is too simplified; RR, R, MS, and S need to be explained. Each figure needs to be explained independently enough on its own.

2.       The same is true for the tables. In Table 2, Final score and AUDPC need to be explained in the notes below the table.

3.       It is unclear what the error bars in the graph in Figure 1 mean. An explanation is needed. Is statistical analysis possible?

4.       The Introduction should provide a detailed introduction to the current situation in Kazakhstan, and the Discussion should emphasize the significance of this study for future wheat breeding in Kazakhstan.

Author Response

REVIEWER 3 – answers in MS in green color

Начало формы

  1. Figure 1 caption is too simplified; RR, R, MS, and S need to be explained. Each figure needs to be explained independently enough on its own.

We agree with your comment. We have added explanations to the pictures. We also improved the materials and methods section where we explained the criteria for assigning infectious types RR, R, MS, and S.

                                                            (a)                                                                          (b)

Figure 1. Summary of infection types among 60 wheat cultivars and breeding lines infected with Z.tritici (a) at the adult plant stage and (b) at the seedling stage. Note: RR – highly resistant; R – resistant; MS – moderately susceptible; S – susceptible.

line 348-355: According to the degree of damage, the varieties were divided into the following groups: 0-10% – highly resistant (RR; free of infection or few isolated lesions on lowest most leaves only); 11-20% – resistant (R; scattered lesions on the second set of leaves with first leaves infected at light intensity); 21-40% – moderately susceptible (MS; moderate to severe infection of lower leaves with scattered to light infection extending to the leaf immediately below the mid-point of the plant); 41-70% – susceptible (S; Lesions severe on lower and middle leaves; moderate to severe infection of upper third of plant; flag leaf infected in amounts more than a trace), 71-100% - highly susceptible (SS; severe infection on all leaves and the spike infected to some degree).

  1. The same is true for the tables. In Table 2, the Final score and AUDPC need to be explained in the notes below the table.

We agree with your comments. The captions under the table have been improved. We have added clarifications to AUDPC and Final Score.

Line 174-175: Final score – latest assessment of plant resistance in the field. AUDPC – area under the disease progress curve.

  1. It is unclear what the error bars in the graph in Figure 1 mean. An explanation is needed. Is statistical analysis possible?

We agree with your comment. We have added the appropriate explanations.  AUDPC and ASR assessment were used to conduct a two-factor analysis of variance and descriptive statistics of wheat genotypes (Table 1 and Table 2).

Line 141-143: The distribution of accessions concerning Z. tritici infection types at the adult plant stage (a) and seedling stage (b) with an indication of the standard error is presented in Figure 1

  1. The Introduction should provide a detailed introduction to the current situation in Kazakhstan, and the Discussion should emphasize the significance of this study for future wheat breeding in Kazakhstan.

We agree with your comment. We ADDED to Introduction information about the current situation in Kazakhstan. Epidemics Zymoseptoria tritici in the North of Kazakhstan occurs 5 times every 10 years. According to the monitoring of Z. tritici in the Akmola region, there is a trend of increased development and severity of the disease, and, in recent years, the morbidity rate has reached a critical level. The disease manifested itself even during the years of hard drought (2003-2010). Strong development of S. tritici was observed in 2013, 2014, and 2016 [5].

Added to Discussion Northern The species composition of pathogens populations of wheat Septoria spot in Northern Kazakhstan in 2018–2019 was comprised of three species of Septoria fungi: Zimoseptoria tritici, Septoria nodorum, and Septoria avenae f. sp. triticea. Soft spring wheat was mainly affected by the S. tritici. During the two-year study of species diversity of Septoria spot pathogens, Z. tritici was predominant followed by S. nodorum and S. avenae f. sp. triticea [5] Babkenova SA, Babkenov AT, Pakholkova EV, Kanafin BK. Pathogenic complexity of septoria spot disease of wheat in northern Kazakhstan. Plant Sci. Today [Internet]. 2020 Oct. 1 [cited 2024 Mar. 3];7(4):601–606. Available from: https://horizonepublishing.com/journals/index.php/PST/article/view/798]. This study has significance for future wheat breeding in Kazakhstan. There is not a single variety resistant to Z. tritici among the varieties approved for use [Babkenova et al., 2020]. As a result of our study, identifying the sources of STB genes with minor-to-moderate effects will evolve as an effective way to have durable and resistant wheat varieties for STB.

Reviewer 4 Report

Comments and Suggestions for Authors

Comments to paper 10.3390 

Title : Identification of wheat septoria tritici resistance gens in wheat germ plasm using molecular markers.

The topic is highly relevant as there needs to be a constant effort in the breeding communities – searching for stable and new sources of resistance to minimize infections from Zymoseptoria tritici.

The work presented seems solid and it is indeed valid to compare seedling test data with field data.

Using SSR marker to identify resistance genes is good and widely used today. The resistance genotypes found in the presented paper can be widely used by the breeding community when they want to strengthen the search for new material.

The outcome from field testing and seedling testing can be very different which confirms that seedling test alone is not robust enough when selectin for new cultivars with good resistance.

Comments to the introduction

Line 42 -43. Be more precise in what is lacking. What means by optimal input application? And what means by lack of appropriate machinery?

Line 53. The word tillage has to be included in this sentence.

Line 56-69. This section is very confusing and need too be better described.

Line 57; septoria  diseases. Should be changed to leaf blotch diseases.

Leaf blotch disease in wheat normally covers 3 diseases. Zymoseptoria tritici, Phaeosphaeria nodorum  and Pyrenophora tritici repentis.

If the Latin name is not used then a consistent English name most be used. Septoria tritici blotch, Septoria nodoum blotch and tan spot respectively.  

67-68: Again the term septoria diseases should not be used. The sentence is confusion – tan spot is part of the leaf blotch diseases.  Sentence should be changed to: Apart from leaf blotch diseases also brown rust, yellow rust and stem rust are widespread.  The first time a disease is mentioned the latin name should also be included in brackets.

We lack a description on how the life cycle of Zymoseptoria tritici is seen in Kazakstan. Does it differ from other regions? In Europe the ascospores are wind spread in the autumn from debris. The spread happens across long distances and debris in the individual field dose not play a role for the severity during the growing season. Pycnidia spores are seen during late winter and spring and from this source of inoculum the disease spread upwards in the canopy.  I notice that you spread out plant debris at the site in the autumn – This we would do to stimulate infection of tan spot but not Septoria tritici blotch.  Do you have evidence that spread of debris help to support attack of STB.

Be consistent in the latin names used. Do not use Mycospharella graminicola (line 238 239) – use in all cases Zymoseptoria tritici.

Material section

Line 368: spore concentration should be 107 not 107. Same should also be adjusted in line 401 which also deals with spore concentrations.

Line 378-382: I get confused reading this paragraph.

10 seeds sown per pot – unit. Each container had 3 seedlings (was 7 removed?). You had 3 replicates (=3 pots?).

But then you write that 10 seedlings was inoculated per treatment at the 2 leaf stage with a mix of Z.tritici isolates. How did you get the 10 seedlings? IF you had 3 seedlings per pot and 3 replicates – it means that you had 9 seedlings in total – not 10 seedlings.

It reads as if you inoculate using one mixture with spores from 5 different isolates which was sprayed on to all genotypes?  Did you mix the 5 isolates before applying them or did you apply them one by one?  In line 405 you write treatment, what do you mean by treatment – expect it to be equal to a genotype/cultivar?

In line 407 you write that you cleaned the spray gun after each treatment?. Why was this- if they all were treated with the same mixture?

Line 417: was each individual seedling rated or did you give an average score on the pot. In other words did you have 3 subsamples per pot or only data from the 3 replicates?   

Results or discussion chapter:

We need also a description of how other diseases (tan spot and septoria nodorum blotch) interfered with the scoring in the field. Did you only see septoria tritici blotch or did other diseases also appear?

Line 217: yield loss. Your mentioned that losses are reaching from 15 to 50%. This is a lot! In many cases yield losses can also be as low as 3-5%.  A min. of 15% sounds like an exaggeration of the importance of the disease.

Discussion

224-226: Not sure I understand this sentence. SSR markers for Stb genes – do they also catch resistance genes for yellow rust?

Line 254:  32 genotypes were lacking Stb genes.  Or is more correct to say that the used SSR markers did not identify Stb genes in 32 genotypes. Could there be problems with some genotypes not responding to the SSR markers – but still having specific Stb genes.  

Discuss the advantages of seedling tests and the draw backs.  Discuss also the limitations in using field testing (influence of other diseases, weather not being conductive, different virulense in natural populations etc. ).

Author Response

Comments to the introduction and Answers. Answers to Reviewer 4 have been added in the manuscript in blue color.

Question Line 42 -43. Be more precise in what is lacking. What means by optimal input application? And what means by lack of appropriate machinery?

Answer Line 42 -43. We agree with the reviewer's comment. Corrected, now the manuscript looks LIKE THIS:

There are several reasons for poor wheat grain yield in the region, including changes in cultural practices including shifts from conventional tillage and stubble burning to reduced tillage practices, breaking of the technology of fertilizing, violation or non-compliance with crop rotation, and shallow tillage and wheat monoculture involving the cultivation of susceptible cultivars. However, diseases and pests also play an important role in yield reduction [2].

Question Line 53. The word tillage has to be included in this sentence.

Answer Line 53. We agree with the reviewer's comment. The word tillage is included in this sentence. Corrected, now the manuscript looks LIKE THIS:

The widespread introduction of zero-tillage and minimal wheat cultivation contributes to the development and harmfulness of leaf spot diseases (LSD), the infection of which remains on crop residues.

Question Line 56-69. This section is very confusing and need too be better described.

Answer Line 56-59. We agree with the comment. We have removed outdated systematic taxa and present only modern ones. To make it clear to the reader. The paragraph now looks LIKE THIS:

In Kazakhstan, 5 species of Septoria fungi were registered on wheat, the dominant among them is Parastagonospora nodorum (Berk.) Quaedvl., Verkley et Crous and Zymoseptoria tritici (Desm.) Quaedvl. et Crous [4]. Septoria tritici blotch (STB) caused by the ascomycete fungus Zimoseptoria tritici, is the major devasting foliar disease that caused significant yield loss in wheat-growing regions of Kazakhstan. Yield losses of grain crops due to these two diseases in the world amount to 9 million tons [5]. The fungus P. nodorum was discovered in 1960 in the Akmola region of Kazakhstan on soft wheat, since then the disease has been observed in all regions of the country. Septoria tritici blotch and Stagonospora nodorum blotch are a major threat to global food security [6]. The average loss from pathogens can reach from 15 to 50% [4]. Leaf blotch diseases are a problem in wheat production in the Pacific Northwest and Northern Great Plains of the USA, Europe, and Central and West Asia [5-6, 7-11]. Apart from leaf blotch diseases also leaf rust (Puccinia triticina Erikss.) [2, 12-13], yellow rust (Puccinia striiformis Westend.) [14-16], and stem rust (Puccinia graminis f. tritici Erikss. & Henning) [17-18] are widespread in Kazakhstan.

Question Line 57; septoria  diseases. Should be changed to leaf blotch diseases.

Leaf blotch disease in wheat normally covers 3 diseases. Zymoseptoria tritici, Phaeosphaeria nodorum  and Pyrenophora tritici repentis.

If the Latin name is not used then a consistent English name most be used. Septoria tritici blotch, Septoria nodoum blotch and tan spot respectively.  

Answer Line 57. We agree with the comment, we have made changes TO THE TEXT:

Leaf blotch diseases are a problem in wheat production in the Pacific Northwest and Northern Great Plains of the USA, Europe, and Central and West Asia [5-6, 7-11]. Apart from leaf blotch diseases also brown rust (Puccinia triticina Erikss.) [2, 12-13], yellow rust (Puccinia striiformis Westend.) [14-16], and stem rust (Puccinia graminis f. tritici Erikss. & Henning) [17-18] are widespread in Kazakhstan.

Question Line 67-68: Again the term septoria diseases should not be used. The sentence is confusion – tan spot is part of the leaf blotch diseases.  Sentence should be changed to: Apart from leaf blotch diseases also brown rust, yellow rust and stem rust are widespread.  The first time a disease is mentioned the latin name should also be included in brackets.

We lack a description on how the life cycle of Zymoseptoria tritici is seen in Kazakhstan. Does it differ from other regions? In Europe the ascospores are wind spread in the autumn from debris. The spread happens across long distances and debris in the individual field dose not play a role for the severity during the growing season. Pycnidia spores are seen during late winter and spring and from this source of inoculum the disease spread upwards in the canopy.  I notice that you spread out plant debris at the site in the autumn – This we would do to stimulate infection of tan spot but not Septoria tritici blotch.  Do you have evidence that spread of debris help to support attack of STB.

Answer Line 67-68. We added to the MS: The life cycle of Zymoseptoria tritici in Kazakhstan is the same as in other regions. The life cycle of Z. tritici is broadly divided into two distinct stages, namely the symptomless latent phase and the necrotrophic stage. The latent phase can be further subdivided into three stages: transition, ingress, and colonization [4]. ВТАВИТЬ только это в ТЕКСТ  стр 2 The fungus overwinters on stubble. Its sexual stage also takes place there. The ascospores then infect winter wheat varieties. Therefore, additional application of crop residues contributes to the accumulation of infectious material on the field. Therefore, additional spread-out plant debris contributes to the accumulation of infectious material on the field. Winter crops are infected with spores of spring wheat varieties. If the winters are warm, then on winter varieties, asexual reproduction of the fungus may occur several times during the winter period with the formation of pycnidia and the spread of spores.

However, since the life cycle of Z. tritici in Kazakhstan is no different from that in other regions, we believe that there is no need to describe this issue in the manuscript itself. We considered it possible not to add detailed information presented in textbooks and classical works to our article. We limited ourselves to a short explanation and a link to the source [4].

Question Line 238-239 Be consistent in the latin names used. Do not use Mycospharella graminicola (line 238 239) – use in all cases Zymoseptoria tritici.

Answer Line 238-239. Agree with the comment. The CHANGES HAVE BEEN MADE:

In different previous studies, the sources of Stb resistance genes were identified in wheat breeding material [32-33]. As a consequence of molecular genetic tagging of resistance genes in 36 varieties of wheat, it was determined that the majority of the analyzed samples carried in their genotype resistance genes ineffective to Z. tritici. The result of a study by Babkenova et al. (2017) showed moderate effective resistance of the Stb2 gene [32]; eight wheat entries with this gene were identified among the studied cultivars. Research conducted by Pakholkova et al., 2016 [33] showed that 6 of the 8 known resistance genes (Stb1-Stb5, Stb7) had only partial functionality in natural populations of Z. tritici in Russia, the Stb6 gene was highly effective against five populations of Z. tritici, and the Stb8 gene was efficient in an absolute sense against all tested isolates. They also, as in the present study, report the usefulness of screening for the presence of Stb genes for the breeding of Septoria-resistant cultivars.

Material section

Question Line 368: spore concentration should be 107 not 107. Same should also be adjusted in line 401 which also deals with spore concentrations.

Answer Line 368. We have made changes and the text looks LIKE THIS:

The spores were collected by centrifugation at 5000 rpm for 5 min at 15 °C, washed twice with sterile distilled water, and resuspended in a MgSO4 solution (0,01 M), containing Tween 20 surfactant (0.1% v/v). The concentration was adjusted to 107 spores/mL [40].

Answer Line 401 The inoculum was adjusted to the required cell density (e.g., 106 or 107 cells/ml) in 0.1% Tween® 20 in sterile water.

Question Line 378-382: I get confused reading this paragraph.

10 seeds sown per pot – unit. Each container had 3 seedlings (was 7 removed?). You had 3 replicates (=3 pots?).

But then you write that 10 seedlings was inoculated per treatment at the 2 leaf stage with a mix of Z.tritici isolates. How did you get the 10 seedlings? IF you had 3 seedlings per pot and 3 replicates – it means that you had 9 seedlings in total – not 10 seedlings.

It reads as if you inoculate using one mixture with spores from 5 different isolates which was sprayed on to all genotypes?  Did you mix the 5 isolates before applying them or did you apply them one by one?  In line 405 you write treatment, what do you mean by treatment – expect it to be equal to a genotype/cultivar?

Answer Line 378-382: Thanks for your note, this is our mistake. The CHANGES HAVE BEEN MADE:

10 seeds of each variety were sown in a pot. We received 10 plant seedlings in a pot. The experiment was repeated three times. Therefore, 30 plants of the same variety were evaluated.

The manuscript has been corrected as follows:

Each container was considered as an experimental unit, and each single plant wheat variety in a container with three ten seedlings served as an entry.

Question: It reads as if you inoculate using one mixture with spores from 5 different isolates which was sprayed on to all genotypes?  Did you mix the 5 isolates before applying them or did you apply them one by one? 

Answer: Yes, the fungal isolates were mixed before inoculation.

We wrote in our MS, Line 345: «Plant material of wheat was inoculated with a mix of Zymoseptoria tritici isolates».

Question Line 405: In line 405 you write treatment, what do you mean by treatment – expect it to be equal to a genotype/cultivar?

Answer Line 405: We meant that for each treatment of plants with a spore suspension. But in our case, there was one treatment with a mix of spores. And we completely agree with the reviewer. Unnecessary information needs to be removed.

The text in MS now looks LIKE THIS:

To inoculate the plants the cleaned spray gun was used and a pressure of 2.0 bar to inoculate the marked leaf sections of each plant with the Z. tritici inoculum by evenly spraying until runoff.

Question In line 407 you write that you cleaned the spray gun after each treatment? Why was this- if they all were treated with the same mixture?

Answer Line 407: We completely agree with the reviewer. Unnecessary information was removed. the text looks LIKE THIS:

To inoculate the plants for each treatment, the cleaned spray gun was used, and a pressure of 2.0 bar to inoculate the marked leaf sections of each plant with the Z. tritici inoculum by evenly spraying until runoff.

Question Line 417: was each individual seedling rated or did you give an average score on the pot? In other words, did you have 3 subsamples per pot or only data from the 3 replicates?   

Answer Line 417: According to the methodological recommendations, we evaluated 10 seedlings separately in each pot. Since we had 3 replicates, we received 30 ratings per variety. Then the results were statistically processed.

 Results or discussion chapter:

We need also a description of how other diseases (tan spot and septoria nodorum blotch) interfered with the scoring in the field. Did you only see septoria tritici blotch or did other diseases also appear?

Question Line 217: yield loss. Your mentioned that losses are reaching from 15 to 50%. This is a lot! In many cases yield losses can also be as low as 3-5%.  A min. of 15% sounds like an exaggeration of the importance of the disease.

Answer Line 217. We agree with the reviewer's comment. Corrected, now the MSl looks LIKE THIS:

The main agricultural crop, wheat, is affected by pathogens of leaf spot diseases (LSD), including Septoria. Z. tritici is the main threat to global food security. Losses due to this disease can reach up to 50% in epidemic years and often vary between 5 and 20% depending on the environment and the cultivar of wheat [4].

 Discussion

Question Line 224-226: Not sure I understand this sentence. SSR markers for Stb genes – do they also catch resistance genes for yellow rust?

Answer Line 224-226. Thanks for your note, this is our mistake. Of course, we meant septoria disease. This sentence has been corrected to look LIKE THIS: Diversity in Stb genes in commercial cultivars could play an important role in managing frequent leaf blotch disease epidemics in the region.

Question Line 254:  32 genotypes were lacking Stb genes.  Or is more correct to say that the used SSR markers did not identify Stb genes in 32 genotypes. Could there be problems with some genotypes not responding to the SSR markers – but still having specific Stb genes.  

Answer Line 254:  Yes, we agree with comment/ We have made changes and the text looks LIKE THIS:

The used SSR markers did not identify Stb genes in 32 genotypes.

Question Discuss the advantages of seedling tests and the drawbacks.  Discuss also the limitations in using field testing (influence of other diseases, weather not being conductive, different virulense in natural populations etc. ).

Answer – We could respond to this comment AS FOLLOWS:

The advantages of seedling tests are: less space and time required; easier to move plants in the greenhouse; fewer problems with greenhouse pests; and minimized risk of contamination by other diseases. The disadvantages of seedling tests are spores produced in small quantities and the frequent need to repeat the process.

When conducting a field evaluation for disease, the study is carried out in natural, close-to-production field conditions in a specially designated area. Although this method also has its Disadvantages this method is the influence of other diseases, weather not being conductive, different virulence in natural populations, etc.  Therefore, the study of seedling and adult plant resistance with the addition of the marker-assisted selection (MAS) approach accelerates the cultivation of new cultivars allowing to minimization of all of the above disadvantages and obtaining maximum disclosure of the genetic potential of each wheat variety.

However, since this information about the advantages and disadvantages of methods for studying disease resistance is well described in classic textbooks, we found it possible not to include this part in the Discussion part of the manuscript.

Round 2

Reviewer 2 Report

Comments and Suggestions for Authors

Authors tried to clarify some of the short coming of the earlier version.  I would like to see further clarification to be considered for publication.

1. Table 1: Still there is not enough information/explanation to explain this.  What is residual?  Do you get 2D output for Z tritici and 1D output for AUDPC?  The current format is extremely difficult to process.  Maybe a better representation would be make a curve for genotype vs (average AUDPC).  

Another questions is how to distinguish the distance among different genotypes.  It looks like they were treated as totally independent variable, but some genotypes are probably highly similar to one another.

2. Figure 1:  It would be beneficial if authors can present the resistance score for 2020 (x axis) and 2021 (y axis) for each genotype.  This will clarify if the quantification is consistent across years.

3. section 4.5: The term "disease assessment" seems subjective.  An example of your assessment as an image would be helpful. 

4. Regarding Stb4 and Stb2: Authors tried to clarify my earlier comment, but still it is not conclusive.  Authors should group samples 1) with one Stb2 (for example) gene and 2) without the Stb2.  If there is significant difference in these two groups (support by a figure and statistics), then such a claim can be made. 

Author Response

  1. Table 1: Still there is not enough information/explanation to explain this.  What is residual?  Do you get 2D output for Z tritici and 1D output for AUDPC?  The current format is extremely difficult to process.  Maybe a better representation would be make a curve for genotype vs (average AUDPC).

Another questions is how to distinguish the distance among different genotypes.  It looks like they were treated as totally independent variable, but some genotypes are probably highly similar to one another.

Thank you for the clarification. We are responding to your question.

Interpreting residuals in the context of two-way analysis of variance is crucial for assessing the adequacy of the model and identifying potential issues. Residuals represent the difference between observed values and values predicted by the model based on the factors considered in the analysis.

The residuals for the AUDPC trait represent the residual variance that remains after accounting for the influence of genotype and year on this indicator. The residuals amount to 46578 with 59 degrees of freedom. This means that after considering the influence of genotype and year on the AUDPC trait, there remains residual variance that is not explained by the factors accounted for.

This is important because residual variance may indicate the presence of other unaccounted factors that could affect the AUDPC trait, or it may indicate the incompleteness of the model.

The residuals for the Z.tritici trait.  represent the residual variance after considering the influence of genotype and replication on this indicator. The residuals amount to 45.63 with 236 degrees of freedom. This also represents residual variance after accounting for the influence of genotype and replication on the Z.tritici trait.

Interpreting residuals helps evaluate how well the model fits the data and how accurate the predictions are. Identifying anomalies in residuals may indicate the need for model adjustment or further data analysis.

In this analysis, we considered the influence of genotypes and year/replication on the resistance indicator (AUDPC/Z.trititci). This table informs us about the influence of genotype as the most significant factor on resistance. Each genotype was considered separately, as all genotypes may inherit various resistance genes (including those not identified in this study). We calculated the heritability index and obtained a result indicating the trait's heritability among genotypes.

The analysis of AUDPC is multidimensional. In this case, years represent one factor, and genotypes represent the second. Thus, in the ANOVA analysis to assess AUDPC, we have two factors: year and genotype. We investigate the influence of both factors on the value of AUDPC.

Based on all of the above, we would like to preserve the results of this analysis in the form of the generated table, as we consider them a classic example of demonstrating the results of two-way analysis of variance.

  1. Figure 1:  It would be beneficial if authors can present the resistance score for 2020 (x axis) and 2021 (y axis) for each genotype.  This will clarify if the quantification is consistent across years.

 We agree with your remarks and believe that this method of presenting the results is informative.

We have prepared Figure 1 to demonstrate the stability over 2 years, with the AUDPC values for 2020 on the x-axis and for 2021 on the y-axis.

Figure 1. The resistance score for Z.tritici with the AUDPC values for 2020 and 2021 for wheat genotypes at the adult plant stage.

To demonstrate consistency in assessment across years, we conducted Pearson correlation analysis and added its results to the publication. A significant positive correlation in resistance values among genotypes over 2 years of field trials was identified (r = 0.94; p < 0.001).

Corrections made Line 140-145: The resistance score with the AUDPC values for 2020 and 2021 for each genotype is presented in Figure 2.  This will clarify if the quantification is consistent across years. This showed if the quantification of resistance is consistent across years. The Pearson correlation analysis revealed a significant positive correlation between the two years of field trials (r = 0.94; p < 0.001). No correlation was found between the resistance of adult plants and seedlings.

  1. section 4.5: The term "disease assessment" seems subjective.  An example of your assessment as an image would be helpful. 

In the assessment of diseases, we used the classical foliar disease rating scale developed by Saari, E.E. and Prescott J.M. (A scale for appraising the foliar intensity of wheat diseases. Plant Dis. Rep. 1975, 59(5), 377-380). Scores ranging from 1 to 9 were converted into percentages for ease of calculation according to the methodology outlined by Eyal, Z., A.L. Scharen, J.M. Prescott, and M. van Ginkel. 1987. The Septoria Diseases of Wheat: Concepts and methods of disease management. Mexico, D.F.: CIMMYT.

10% coverage = 1
20% coverage = 2
30% coverage = 3
            ---
90% coverage = 9

We also prefer not to include this image in the publication, as a reference to the methodology is provided.

  1. Regarding Stb4 and Stb2: Authors tried to clarify my earlier comment, but still it is not conclusive.  Authors should group samples 1) with one Stb2 (for example) gene and 2) without the Stb2.  If there is significant difference in these two groups (support by a figure and statistics), then such a claim can be made. 

Lines 256-276 (Results) Regarding Stb genes: An analysis of the general distribution of dependent variables of various wheat samples characterized by the presence and absence of Z.trititci resistance genes Stb2, Stb4, Stb7, and Stb8 was conducted (Supplementary Figures 1-4). For this purpose, the effectiveness of carriers and non-carriers of Stb2, Stb4, Stb7, and Stb8 genes for the Z.trititci score was compared using a non-parametric analog of the analysis of variance and the Kruskal-Wallis criterion.

            As suggested by the reviewer, we have grouped the samples into two categories: 1) those with one Stb2 gene and 2) those without the Stb2 gene. In these groups, no significant difference (P > 0.05) was observed at either the seedling or adult plant stages (Supplementary Figures 1). The analysis revealed that there were no statistically significant differences between the carrier and non-carrier groups of other genes (Stb4, Stb7, and Stb8). This is indicated by high p-values (P > 0.05) for both seedling and adult plants (Supplementary Figures 2-4).
Among the studied Stb genes, the Stb8 gene demonstrated the greatest influence on resistance to Z.trititci, as statistical analysis showed that the lowest p-value (0.19) was observed between carriers and non-carriers of the Stb8 gene (Supplementary Figures 4). Hence, the presence of this gene in wheat genotypes significantly enhanced resistance to Z.trititci.

However, the significant impact of the year factor on resistance was also observed (P < 0.05). Supplementary Figure 5-10 illustrates the level of Z.trititci development in various wheat groups under field conditions across two years (2020 and 2021). The average disease development level in 2020 ranged from 11.4% to 11.8%, whereas in 2021, this figure increased from 15.7% to 16.6% (Supplementary Figure 5-10).

Lines 349-355 (Discussion) Thus, it was found that the factor of the year significantly influences the development of Z.trititci  in field conditions between carrier and non-carrier groups of Stb2, Stb4, Stb7, and Stb8 genes. This indicates that the influence of genetic factors may vary depending on the conditions of the year. The average level of Z.trititci  in 2021 is higher than in 2020 for both carriers and non-carriers of Stb resistance genes. This may suggest an increase in the infection rate in the latter year, regardless of the presence or absence of specific genetic variants.

A

Min.

1.0    

1st Qu. 

2.2    

Median

2.6    

Mean

2.4    

3rd Qu. 

3.0    

Max.

3.6

B

Stb genes

Z.tritici (seedling)

Stb2_carriers

2,2

non_Stb2_carriers

2,5

P value

0.37

Supplementary Figure 1. The general distribution of the Stb2 variable (A) and the boxplot for the Stb2 variable (B)

.

A

Min.

1.0  

1st Qu. 

1.6  

Median

2.4     

Mean

2.3    

3rd Qu. 

3.0      

Max.

3.8

B

Stb genes

Z.tritici (seedling)

Stb4_carriers

2,2

non_Stb4_carriers

2,3

P value

0.51

Supplementary Figure 2. The general distribution of the Stb4 variable (A) and the boxplot for the Stb4 variable (B)

Min.

1.0  

1st Qu. 

   1.6   

Median

2.4    

Mean

2.3  

3rd Qu. 

3.0     

Max.

3.8

Stb genes

Z.tritici (seedling)

Stb7_carriers

2.28

non_Stb7_carriers

2,26

P value

0.62

Supplementary Figure 3. The general distribution of the Stb7 variable (A) and the boxplot for the Stb7variable (B)

Min.

  1.0  

1st Qu. 

1.6  

Median

2.4  

Mean

2.3  

3rd Qu. 

3.0  

Max.

3.8

Stb genes

Z.tritici (seedling)

Stb8_carriers

2.52

non_Stb8_carriers

2.16

P value

0.19

Supplementary Figure 4. The general distribution of the Stb8 variable (A) and the boxplot for the Stb8 variable (B)

A

Min.

0.0

1st Qu. 

6.2

Median

10.0

Mean

13.5

3rd Qu. 

15.0

Max.

50.0

B

Min.

0.0

1st Qu. 

5.0

Median

10.0

Mean

14.1

3rd Qu. 

15.0

Max.

50.0

Supplementary Figure 5. General distribution of Stb2 (A) and Stb4 (B) disease scores in the field

A

Min.

0.0

1st Qu. 

5.0

Median

10.0

Mean

14.1

3rd Qu. 

15.0

Max.

60.0

B

Min.

0.0

1st Qu. 

5.0

Median

10.0

Mean

14.1

3rd Qu. 

15.0

Max.

6.0

Supplementary Figure 6. General distribution of Stb7 (A) and Stb8 (B) disease scores in the field

Stb genes

Z.tritici (field)

Stb2_carriers

15.7

non_Stb2_carriers

12.4

P value

0.42

Year

Z.tritici (field)

2020

11.4

2021

15.7

P value

0.01

Supplementary Figure 7. Boxplots of Stb2 and year disease scores (2020-2021) in the field

Stb genes

Z.tritici (field)

Stb4_carriers

11.1

non_Stb4_carriers

15.2

P value

0.32

Year

Z.tritici (field)

2020

11.8

2021

16.4

P value

<0.01

Supplementary Figure 8. Boxplots of Stb4 and year disease scores (2020-2021) in the field

Stb genes

Z.tritici (field)

Stb7_carriers

9,2

non_Stb7_carriers

15.2

P value

0.11

Year

Z.tritici (field)

2020

11.8

2021

16.4

P value

<0.01

Supplementary Figure 9. Boxplots of Stb7 and year disease scores (2020-2021) in the field

Stb genes

Z.tritici (field)

Stb8_carriers

16.1

non_Stb8_carriers

13.1

P value

0.16

Year

Z.tritici (field)

2020

11.8

2021

16.6

P value

<0.01

Supplementary Figure 10. Boxplots of Stb8 and year disease scores (2020-2021) in the field

.

Round 3

Reviewer 2 Report

Comments and Suggestions for Authors

The manuscript has improved a bit, but still there is no data to support authors' claim that Stb genes are key to resistance. 

Please, note that presence of Stb genes does not mean causality.  It is highly possible that Stb genes was maintained without any consequence of resistance. 

I am ok to report that Stb genes are found in certain cultivars, but the connection between Stb genes and resistance is not supported by data provided.

Author Response

Answer to Reviewer 2 – 04-04-2024

  1. 1. The manuscript has improved a bit, but still there is no data to support authors' claim that Stb genes are key to resistance. 

Lines 277-285. In response to your question about the lack of data supporting the authors' claim that Stb genes are key to resistance, we conducted a comparative study on field resistance of wheat samples. Positive samples containing Stb genes were compared with the resistance of negative control plants of the variety Morocco, which lacked Stb genes. Statistical parameters were assessed, including 162 measurements for positive and 162 measurements for negative samples concerning the presence of Stb genes, respectively. The statistical parameters for positive samples ranged within Min.- 0.0, 1st Qu.- 5.0, Median – 5.0, Mean – 8.9, 3rd Qu.- 10.0, Max. -50.0. For negative samples, they increased by 5-7 times to Min.- 30.0, 1st Qu.- 50.0, Median – 55.0, Mean – 53.8, 3rd Qu.- 60.0, Max. -65.0 (Supplementary Figures 11-12).

Lines 286-293. A significant influence of factors year and genetic background (positive and negative samples concerning the presence of Stb genes) on Z. tritici severity at the adult plant stage (field studies) was detected (P value <0.01) (Supplementary Figures 13-14). The average severity of Septoria leaf blotch in 2020 and 2021 was 32.21% and 37.21%, respectively (Supplementary Figure 13). The average severity of Septoria leaf blotch by genetic backgrounds for positive samples (Stb gene carriers) ranged within 8.51% in comparison, for negative samples (lack of Stb genes) increased by 6 times to 53.8% (Supplementary Figure 14).

Min.

1st Qu

Median

Mean

3rd Qu.

Max.

  0.000  

5.000  

5.000  

8.889 

10.000 

50.000    

Supplementary Figure 11 – The General distribution of overall severity of septoria leaf blotch in positive wheat samples concerning the presence of Stb genes

Min.

1st Qu

Median

Mean

3rd Qu.

Max.

30.0   

   50.0   

55.0   

53.8   

60.0   

65.0    

Supplementary Figure 12 – The general distribution of overall severity of septoria leaf blotch in negative wheat samples concerning the presence of Stb genes

year

Septoria score

2020

32,21

2021

37.21

P value

<0.01

Supplementary Figure 13 - Boxplot showing the influence of the year factor on septoria leaf blotch severity in wheat samples with Stb genes (2020-2021)

Начало формы

Background

Septoria score

Negative

53,80

Positiva

8.51

P value

<0.01

Supplementary Figure 14 - Boxplot showing the influence of the genetic background factor (positive and negative wheat samples with respect to the presence of Stb genes on Z. tritici severity.

  1. Please, note that presence of Stb genes does not mean causality.  It is highly possible that Stb genes was maintained without any consequence of resistance. 

Lines 294-300. Similar calculations were conducted to obtain data confirming that Stb genes influence resistance at the seedling stage. However, no influence of the genetic background factor (positive and negative samples concerning the presence of Stb genes) on Z. tritici severity at the seedling stage was found (P value <0.9). This suggests, as the reviewer indicated, that the presence of Stb genes does not necessarily imply a causal relationship. Our statistical analyses demonstrated a significant influence of the presence of Stb genes on enhancing resistance only at the adult plant stage (under field conditions).

  1. I am ok to report that Stb genes are found in certain cultivars, but the connection between Stb genes and resistance is not supported by data provided.

Lines 301-304. Thus, Stb genes have been identified in 27 varieties. Our analyses suggest that the correlation between the presence of Stb genes in these 27 varieties and resistance is supported by the provided field resistance data, but not supported by data on seedling resistance.

Round 4

Reviewer 2 Report

Comments and Suggestions for Authors

The new supplemental figures 11/12/14 provides valuable insights on the function of Stb genes. 

1. A detailed description of blotch severity and samples needs to be added.

2. The nature of 162 measurements are the key to support author's claim.  The relevance of Stb genes can be tested only if multiple independent positive and negative Stb-carrying geneopyes are sampled.  If 162 samples were measured from one particular genopyte of Stb-positive/negative genotype, it does not mean causality. 

Author Response

The new supplemental figures 11/12/14 provides valuable insights on the function of Stb genes. 

  1. A detailed description of blotch severity and samples needs to be added.

Answer 1. We agree with the comment. The results of the statistical analysis have allowed for a more detailed description of Z. tritici severity in the following groups: 1) Stb-positive genotypes (27 wheat genotypes carrying Stb, 162 measurements) and 2) Stb-negative genotypes (9 genotypes, 55 measurements lacking Stb genes). Below, in response to the second comment, we have added this data.

  1. The nature of 162 measurements is the key to support author's claim.  The relevance of Stb genes can be tested only if multiple independent positive and negative Stb-carrying genotypes are sampled. If 162 samples were measured from one particular genotype of Stb-positive/negative genotype, it does not mean causality.  

Answer 2. Thank you for your valuable comment. You are correct, and we will certainly take your suggestions and comments into account in our future work. Line 277-297. To assess the influence of Stb genes on resistance to Z. tritici, we compared the field resistance levels of wheat accessions categorized into two groups: those carrying Stb genes (referred to as the positive group) and those lacking Stb genes (referred to as the negative group). We conducted the statistical analysis using datasets representing a single variable, which comprised two distinct levels: 1) the Stb-positive group, consisting of 27 wheat genotypes with Stb genes (totaling 162 measurements), and 2) the Stb-negative group, comprising 9 genotypes without Stb genes (with 55 measurements). The general variable analysis of the variable indicated a population distribution between the groups falls within the range of Min. - 0.0; 1st Qu and Median – 10; Mean – 18.7; 3rd Qu. – 25.0; Max. – 65.0 (Supplementary Figure 11).

The average severity of Z. tritici in 2020 and 2021 was recorded as 32.21% and 37.21%, respectively (Figure 3). In the positive group (Stb gene carriers), the mean severity of Z. tritici was 16.25%, while in the negative group (lacking Stb genes), it was higher at 20.25%. A statistically significant difference was observed between the two genotype groups (P value <0.01).

In the positive genotype group, 19% of plants exhibited immunity to the disease (0% severity); the majority of plants in this group (75%) demonstrated resistance to Z. tritici; the remaining 25% of plants were susceptible to Z. tritici. In the negative genotype group, no wheat varieties exhibited immunity; 29% of plants showed moderate susceptibility, while 71% of plants were susceptible to Z. tritici (Supplementary Figure 11). Thus, a significant influence of genetic background (positive and negative samples concerning the presence of Stb genes) on Z. tritici severity at the adult plant stage (field studies) was detected (P value <0.01) (Supplementary Figures 11).

Supplementary Figure 11. The general distribution of variable severity of Z.tritici in positive

and negative wheat samples concerning the presence/absence of Stb genes

positive and negative wheat samples concerning the presence of Stb genes on Z. tritici severity

   Min.

1st Qu

Median

Mean

3rd Qu.   

Max.

0.0   

10.0   

10.0   

18.7   

25.0   

65.0

Background

Z. tritici score, %

Negative

20,25

Positive

16,25

P value

<0.01

Detailed description of Z. tritici severity of wheat genotypes in positive (27 cvs) and negative (9 cvs) groups

Z. tritici score, %

Resistant groups

Number of Positive plants (%)

Number of Negative plants (%)

0

RR

19

0

5

RR

23

0

10

RR

22

17

15

R

6

5

20

R

5

7

25

MS

8

23

30

MS

1

1

45

S

16

47

Average disease score (P value <0.01)

20,25

16,25

Notes: According to the degree of damage to Z. tritici, the varieties were divided into the following groups: 0-10% – highly resistant (RR); 11-20% – resistant (R); 21-40% – moderately susceptible (MS); 41-100% – susceptible (S).

Round 5

Reviewer 2 Report

Comments and Suggestions for Authors

New explanation is strong, and supports authors' claim.